# Nanostructuring one-dimensional and amorphous lithium peroxide for high round-trip efficiency in lithium-oxygen batteries

Arghya Dutta [1,2], Raymond A. Wong[1,2,3], Woonghyeon Park[4], Keisuke Yamanaka[5], Toshiaki Ohta[5], Yousung Jung [4,6] & Hye Ryung Byon[1,2,6]

The major challenge facing lithium–oxygen batteries is the insulating and bulk lithium peroxide discharge product, which causes sluggish decomposition and increasing overpotential during recharge. Here, we demonstrate an improved round-trip efficiency of ~80% by means of a mesoporous carbon electrode, which directs the growth of one-dimensional and amorphous lithium peroxide. Morphologically, the one-dimensional nanostructures with small volume and high surface show improved charge transport and promote delithiation (lithium ion dissolution) during recharge and thus plays a critical role in the facile decomposition of lithium peroxide. Thermodynamically, density functional calculations reveal that disordered geometric arrangements of the surface atoms in the amorphous structure lead to weaker binding of the key reaction intermediate lithium superoxide, yielding smaller oxygen reduction and evolution overpotentials compared to the crystalline surface. This study suggests a strategy to enhance the decomposition rate of lithium peroxide by exploiting the size and shape of one-dimensional nanostructured lithium peroxide.

[1] Department of Chemistry, Korea Advanced Institute of Science and Technology (KAIST), 291 Daehak-ro, Yuseong-gu, Daejeon 34141, Republic of Korea. [2] Byon Initiative Research Unit, RIKEN, 2-1 Hirosawa, Wako, Saitama 351-0198, Japan. [3] Department of Energy Sciences, Tokyo Institute of Technology, 4259 Nagatsuta-cho, Midori-ku, Yokohama 226-8502, Japan. [4] Graduate School of Energy, Environment, Water and Sustainability (EEWS), KAIST, 291 Daehak-ro, Yuseong-gu, Daejeon 34141, Republic of Korea. [5] Synchrotron Radiation (SR) Center, Ritsumeikan University, Kusatsu, Shiga 525-8577, Japan. [6] Advanced Battery Center, KAIST Institute for NanoCentury, 291 Daehak-ro, Yuseong-gu, Daejeon 34141, Republic of Korea. Correspondence and requests for materials should be addressed to Y.J. (email: ysjn@kaist.ac.kr) or to H.R.B. (email: hrbyon@kaist.ac.kr)

The limitation of driving distance per charge is one of the core challenges associated with electric vehicles in supplanting fossil fuel-powered and environmentally unfriendly vehicles[1,2]. Current battery technology supplies far lower gravimetric and volumetric energy densities compared to fossil fuels, which has propelled the development of advanced batteries[3]. In this regard, the rechargeable lithium–oxygen (Li–O$_2$) battery is one of the most suitable concepts with the essential precondition of high theoretical energy density (~3 kWh kg$^{-1}$). In non-aqueous Li–O$_2$ cells, the O$_2$ electrochemistry consists of an overall two-electron transfer ($2e^-$) producing solid lithium peroxide (Li$_2$O$_2$) during discharge (DC), which subsequently decomposes in the reverse reaction during recharge (RC), i.e., $2Li^+ + O_2 + 2e^- \leftrightarrows Li_2O_2(s)$[4,5]. The low molecular weight of reactants, i.e., Li$^+$ and O$_2$ gas, and lightweight carbon electrode results in high specific capacities, which are typically over 1000 mAh g$^{-1}$$_{electrode}$. However, the bulk Li$_2$O$_2$ produced from DC leads to sluggish decomposition due to its insulating wide bandgap[6], which is reflected in the high overpotential ($\eta$, the difference between thermodynamic reversible potential and measured potential) during RC. Moreover, when the RC potential is over 3.75 V, instabilities of the carbon electrode and non-aqueous electrolyte are exacerbated[7–9]. Therefore, promoting Li$_2$O$_2$ decomposition at little expense of overpotential is highly necessary, which has led to the concerted effort in developing catalysts[10,11] including heterogeneous[12–14] and soluble molecular catalysts[15–17]. These catalysts have shown suppressed RC potentials, but they have also caused unintended problems, such as the degradation of electrolyte solution[18] and shuttling of soluble molecules that passivate the negative Li electrode[19,20].

Alternatively, we have focused on controlling Li$_2$O$_2$ morphology and structure to enable its facile decomposition. We demonstrated the formation of amorphous and film-like Li$_2$O$_2$ in contrast to crystalline and aggregated (or toroidal) Li$_2$O$_2$ particles by tailoring the carbon electrode surface with oxygen moieties[21]. The significant advantage of amorphous Li$_2$O$_2$ is the far higher ionic conductivity and improved electrical conductivity in comparison to crystalline Li$_2$O$_2$[22]. With amorphous structure, Li$_2$O$_2$ facilely decomposes from the surface, corresponding to the Li$_2$O$_2$/electrolyte solution interface at potentials below 3.5 V[21,23,24]. However, such behavior is only observed at the initial RC region, up to ~30% state of RC[21], and is followed by a sudden increase in potential, signifying the slow decomposition of the subsequent process. The high electrical resistivity of bulk Li$_2$O$_2$ may cause sluggish charge transport and requires over 4.0 V for full decomposition[24]. Considering all these processes, the short-lived effect of surface decomposition can largely be attributed to the limited surface area of the Li$_2$O$_2$ film that is accessible to the electrolyte solution. Therefore, a significant increase in the surface area of Li$_2$O$_2$ can promote the decomposition at low overpotential and circumvent the sluggish charge transport.

Here, we demonstrate one-dimensional (1-D) nanostructures of amorphous Li$_2$O$_2$ that can remarkably reduce the RC potential and lead to the high round-trip efficiency of 80%. This result could be achieved by the use of a mesoporous carbon electrode, which guides the growth of these 1-D Li$_2$O$_2$ nanostructures. The fast decomposition of the 1-D nanostructures occurs even at the high RC current rate of 2 A g$^{-1}$$_{carbon}$, which is notably distinct from typical Li$_2$O$_2$ films and large toroidal particles.

## Results

**Discharge–recharge profiles of Li–O$_2$ cells.** As the mesoporous carbon electrode in Li–O$_2$ cells, CMK-3 was employed without the use of any additive carbon. CMK-3 has a turbostratic structure comprised of three-dimensional (3-D) hexagonal arrays

of mesoporous channels and micropores (Supplementary Figure 1)[25]. The average diameters of the mesoporous channels ($d_{meso}$) and micropores ($d_{micro}$) are 3.74 nm and 0.71 nm, respectively, which contribute to the high surface area (SA$_{total}$) of 1128 m$^2$ g$^{-1}$ and large volume ($V_{total}$) of 0.9 cm$^3$ g$^{-1}$ (N$_2$ adsorption–desorption isotherm in Supplementary Figure 1 and Supplementary Table 1). To compare the Li–O$_2$ cell performance of the CMK-3 electrode, we also prepared three non-mesoporous carbon electrodes consisting of large porous carbon (LPC, pore diameter of 80 nm), commercial carbon of Ketjen Black (KB), and multi-walled carbon nanotubes (CNTs), respectively. Figure 1a shows the DC–RC profiles of Li–O$_2$ cells with the above-mentioned carbon electrodes under the same operating conditions. At a glance, all carbon electrodes show comparable DC potentials, but exhibit significantly different RC profiles. At ~67% of the RC process ($Q/Q_{total}$, denoted with the notation 0.67RC) corresponding to 1.0 mAh, the potential for CMK-3 is limited to 3.45 V, while in case of all non-mesoporous carbon electrodes, the potential reaches 4.1 V. Furthermore, CMK-3 shows even lower RC potential in comparison to the catalyst-containing electrodes. At 1.0 mAh, the representative catalysts of cobalt oxide (Co$_3$O$_4$) and ruthenium (Ru) nanoparticles-loaded CNT electrodes attain 3.75–3.8 V, and the soluble redox mediator of TEMPO ((2,2,6,6-tetramethylpiperidin-1-yl)oxyl) reaches 3.6 V. Overall, the CMK-3 electrode exhibits the highest round-trip efficiency (~80%), which is 10–13% higher than the non-mesoporous carbon electrodes and 4–7% higher than the catalyst-containing electrodes (Fig. 1b). These results underscore that CMK-3 itself is

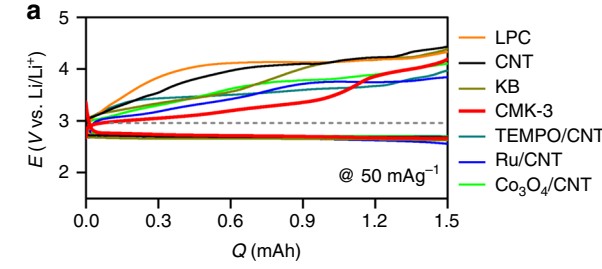

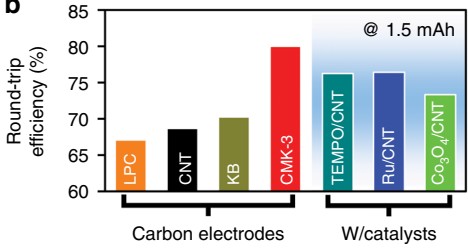

**Fig. 1** Galvanostatic cycle. **a** Discharge (DC)–recharge (RC) profiles for the first cycle with mesoporous carbon (CMK-3, red), ketjen black nanoparticles (KB, dark yellow), carbon nanotube (CNT, black), and large porous carbon (LPC, pore diameter ≈80 nm, orange). All electrodes have same carbon mass of ~1.0 mg and galvanostatic tests were performed at a same current rate of 50 mA g$^{-1}$$_{carbon}$ in 0.5 M LiTFSI in tetraglyme to the fixed capacity of 1.5 mAh. These controlled conditions allow us to assess the round-trip efficiencies. The dashed line indicates a reversible potential of Li–O$_2$ electrochemistry (2.96 V). Along with various carbon electrodes, DC–RC curves with heterogeneous catalysts such as Ru and Co$_3$O$_4$ nanoparticles with CNT (Ru/CNT, blue and Co$_3$O$_4$/CNT, green) and homogeneous catalyst of 10 mM TEMPO ((2,2,6,6-tetramethylpiperidin-1-yl)oxyl) (dark cyan) with CNT were performed. **b** Comparison of round-trip efficiency calculated from the ratio of the integrated areas under the DC and RC curves

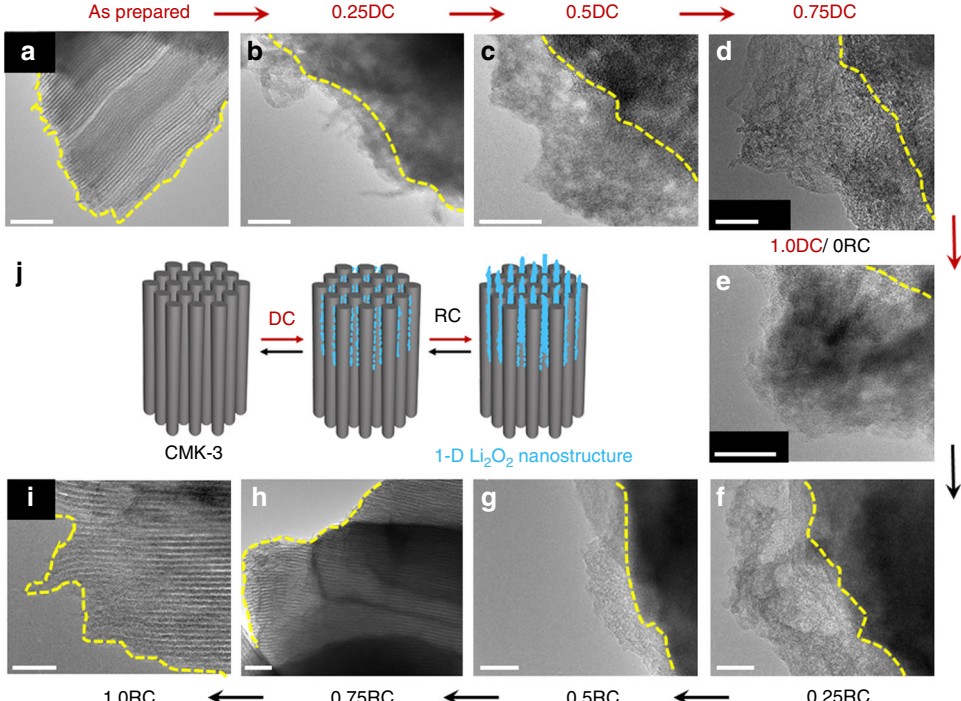

**Fig. 2** Transmission electron microscopy (TEM) images. **a** As-prepared, **b–e** discharging, and **f–i** recharging CMK-3 surfaces. The scale bars are 50 nm. The top label of the image denotes the depth of DC or RC, denoted by $Q/Q_{total}$ at a fixed $Q_{total}$ of 1.0 mAh and the current rate of 50 mA $g^{-1}_{carbon}$. The yellow dashed lines indicate the surface of CMK-3. **j** Schematic illustration of initial growth of $Li_2O_2$, blue poducts, from CMK-3 electrode. From over 0.5DC, the one-dimensional structure and ultrathin $Li_2O_2$ grow upward by loosely entangling with one another, which display flake-like shapes (see SEM images in Supplementary Figure 4)

profoundly effective in suppressing the rise of RC potential in Li–O₂ cells.

**Morphological and chemical characterizations of DC product.** We can infer that the differences in the RC potential profiles are correlated with the morphological and structural characteristics of the DC products. Scanning electron microscopy (SEM) and X-ray diffraction (XRD) reveal film or large particle-shaped products and crystalline $Li_2O_2$ reflections present on the non-mesoporous carbon electrodes of LPC, KB, and CNT following DC (Supplementary Figures 2–3). In the presence of $Co_3O_4$ and Ru nanoparticles (NPs), the crystalline reflections of $Li_2O_2$ are suppressed (Supplementary Figure 3), and the resulting poorly crystalline $Li_2O_2$ is one of the key reasons for the lower RC potential in comparison to the catalyst-free CNT electrodes (Fig. 1a), as we demonstrated in previous reports[13,24]. In stark contrast, flake-like products appear on the CMK-3 electrode surface from SEM images (Supplementary Figure 4). The higher-resolution imaging tool of transmission electron microscopy (TEM) addresses the detailed morphology as ultrathin 1-D nanostructures with a width of 6–15 nm (Fig. 2a, b). They grow upward by loosely entangling with one another during DC (from 0.25DC to 1.0DC, Fig. 2b–e) and their contours display flake-like shapes at 1.0DC (Fig. 2e). There is absence of crystalline $Li_2O_2$ reflections from the XRD analysis (Supplementary Figure 3), revealing its amorphous character, while the DC products can be identified as $Li_2O_2$ from X-ray absorption near-edge structure (XANES) spectroscopy. The O K-edge XANES spectra demonstrate the notable σ*(O–O) bands at 530.5 eV arising from $Li_2O_2$[26] in both total electron yield (TEY) and partial fluorescence yield (PFY) modes (Fig. 3a), which has the escape depth of tens and hundreds of nanometers, respectively. This result indicates

the presence of $Li_2O_2$ on both the surface and bulk (interior) of mesoporous CMK-3. Fourier transform infrared (FTIR) spectra also reveal the Li–O stretching mode at 535 $cm^{-1}$ from $Li_2O_2$ (Supplementary Figure 5)[27]. Although additional vibration bands of side products such as lithium carboxylates and carbonates are also present in the FTIR spectra, the predominant DC product is $Li_2O_2$, and is further confirmed from the average electrons per oxygen molecule of 2.06 $e^-/O_2$, acquired by monitoring the pressure change in the Li–O₂ cell during DC (Fig. 3b, c). Taken together, the DC product formed with CMK-3 has 1-D shape at the nanometer scale and amorphous structure, which is considerably different from typical $Li_2O_2$ films.

**Growth mechanism of $Li_2O_2$ and density functional theory calculations.** The 1-D nanostructured $Li_2O_2$ appearing from the electrode mimics the shape of the mesoporous channels of CMK-3, which is not found with the other non-mesoporous electrodes (Supplementary Figure 2). This allows us to infer that the mesoporous structure of CMK-3 determines the growth process and overall features of the 1-D $Li_2O_2$. With CMK-3, the $O_2$ gas and electrolyte solution diffuse into the mesoporous channels by capillary force[28,29], where the confined $O_2$ is reduced via two-electron transfer as follows:

$$Li^+ + e^- + O_2^* \leftrightarrows LiO_2, \quad (1)$$

$$Li^+ + e^- + LiO_2^* \leftrightarrows Li_2O_2(s). \quad (2)$$

Here, * represents adsorbed species on the surface. After Eq. (1), $Li_2O_2$ can form by either Eq. (2) or by the disproportionation of $LiO_2$ species ($2LiO_2 \leftrightarrows Li_2O_2(s) + O_2$). At the initial stage of DC with CMK-3, we found that the overpotential of oxygen reduction reaction (ORR), which leads to the formation of amorphous

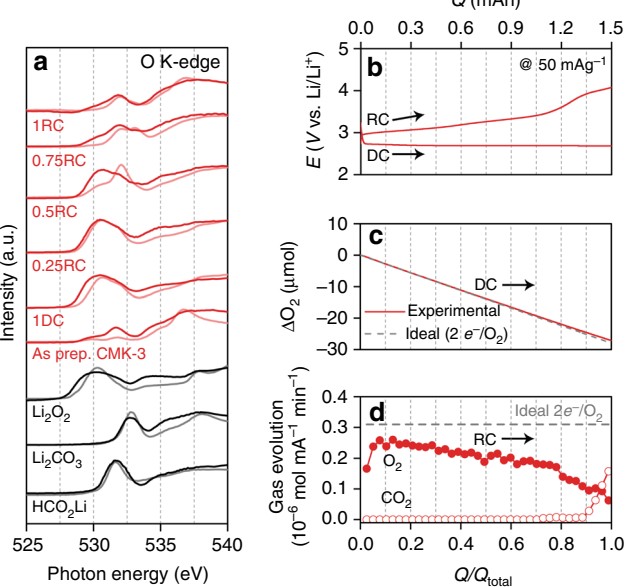

**Fig. 3** Monitoring DC and RC processes. **a** O K-edge XANES spectra for CMK-3 electrode (red) at different depths of DC/RC and standard references of $Li_2O_2$, $Li_2CO_3$, and $HCO_2Li$ (black). The dark and light colors indicate bulk-sensitive partial fluorescence yield (PFY) and surface-sensitive total electron yield (TEY) modes, respectively. The different DC and RC states are denoted from $Q/Q_{total}$ at a fixed $Q_{total}$ of 1.0 mAh and a current rate of 50 mA g$^{-1}_{carbon}$. **b–d** In situ pressure and gas analyses during DC and RC, respectively, using online electrochemical mass spectrometry (OEMS): **b** DC–RC curve at a current rate of 50 mA g$^{-1}$ and a capacity of 1.5 mAh. **c** The gas-pressure decrease during DC (overall 2.06$e^-$/$O_2$). **d** $O_2$ gas (overall 3.17$e^-$/$O_2$) and $CO_2$ gas (over 0.9RC) evolution with the corresponding RC. The horizontal dashed line represents the ideal 2$e^-$/$O_2$

$Li_2O_2$ in the mesoporous channels, was lower than the crystalline $Li_2O_2$ formed with the non-mesoporous electrodes (Supplementary Figure 6). The lower ORR overpotential on the amorphous $Li_2O_2$ surface can be understood by using density functional theory (DFT) calculations to examine the key intermediates of the electrochemical reactions.

Figure 4a, b shows the calculated free energy diagrams along with the optimized structures of crystalline and amorphous surface, respectively. Three different applied potentials ($U$) are incorporated into the energetics via $-neU$, where $n$ is the number of transferred ($Li^+ + e^-$). The overpotential ($\eta$) for ORR is defined as $\eta_{ORR} = U_0 - U_{DC}$, where the equilibrium potential $U_0$ is defined as the potential for which the change of free energy for the whole process is zero, and the DC potential $U_{DC}$ is the highest potential that makes the free energy for every step in ORR downhill.[30,31] Using these prescriptions, the calculated $U_0 = 2.77$ V for the crystalline structure, reconstructed $Li_2O_2$ ($1\bar{1}00$) that is the most stable surface[30], is in good agreement with the experimental thermodynamic potential of $Li_2O_2$ formation (2.96 V).[32] For the amorphous $Li_2O_2$ surface, we divided the surface into 16 equivalent areas to find the lowest energy adsorption site (Supplementary Figure 7), and the free energy diagram shown in Fig. 4b is based on the most favorable reaction site. We note, as summarized in Supplementary Table 2, that statistically 14 sites out of 16 show lower $\eta_{ORR}$ compared to that from the crystalline surface, which is in accordance with the experimental results (Supplementary Figure 6). The potential determining step (PDS) for ORR is calculated to be Eq. (2) for both amorphous and crystalline surfaces, indicating that the weaker binding of second *$LiO_2$ adsorption on the amorphous surface is responsible for a

lower $\eta_{ORR}$. Structurally, this weaker binding of *$LiO_2$ on the amorphous surface is due to the disordered arrangements of the surface Li and O atoms, preventing the newly adsorbing *$LiO_2$ species from forming optimal coordination with the surface. To quantify this, in Fig. 4f, we plotted the binding energies of *$LiO_2$ for the PDS of ORR as a function of the number of newly formed Li–O bonds as a result of *$LiO_2$ adsorption. Overall, the low *$LiO_2$ binding energies on the amorphous surface are indeed associated with the low coordination number that the newly adsorbed *$LiO_2$ forms with the surface, compared to that on the crystalline surface.

On the basis of the DFT results, we expect the weakly binding $LiO_2$ to constitute a chelating network within the confined channels[33,34], prior to the formation of amorphous $Li_2O_2$. As the DC proceeds at low DC current rate, amorphous $Li_2O_2$ grows and emerges from the mesoporous channel. The continuous 1-D growth is also observed on the electrode surface as shown in Fig. 2a–e, j, and is similar in shape to the mesoporous channels. Although the prominent vertical growth is apparent, $LiO_2$ species may also bind at the lateral sides of the 1-D $Li_2O_2$ protruding from the exterior of the mesoporous channels. This can account for the slightly increased width of the nanostructure than the diameter of mesoporous channel. It is noted that the mesoporous channels are only partially filled with $Li_2O_2$ due to slow $O_2$ diffusion and rapid clogging by $Li_2O_2$[29]. This is verified from the full DC capacity of ~2300 mAh g$^{-1}_{CMK-3}$ that is lower than the theoretical capacity of ~2430 mAh g$^{-1}_{CMK-3}$ estimated from the full volume of the mesoporous channels, despite the inclusion of $Li_2O_2$ formed over the exterior of the channels (Supplementary Figure 8). Therefore, the initial growth occurring in the vicinity of channel entry is critical to lead to the unique shape of $Li_2O_2$.

**Analysis of RC process.** We now turn our attention to the RC process of the 1-D amorphous $Li_2O_2$ occuring at low RC potentials. The decreasing height of 1-D nanostructured $Li_2O_2$ is followed by its complete depletion from the CMK-3 surface at 0.75RC as shown by TEM images (Fig. 2f–i). The surface-sensitive TEY mode from the O K-edge XANES spectra is also consistent with this result, showing the significant decrease in the $\sigma^*$(O–O) band by 0.5RC and disappearance at 0.75RC (light solid line in Fig. 3a). However, regarding the interior of CMK-3, the behavior of $Li_2O_2$ decomposition is different. The bulk-sensitive PFY mode shows the pronounced $\sigma^*$(O–O) band at 0.5RC, which decreases at 0.75RC and disappears at 1RC (dark solid line in Fig. 3a). These results clearly demonstrate the preferential decomposition of $Li_2O_2$ on the exterior of CMK-3, relative to $Li_2O_2$ in the interior of the mesoporous channels, which we will discuss in more detail below. Further insight on the decomposition process is gained from in situ gas analysis using on-line electrochemical mass spectrometry (OEMS). Initially, $O_2$ gas evolves at the significant rate of ~0.25 μmol mA$^{-1}$ min$^{-1}$, but gradually decreases throughout the RC process (Fig. 3b, d). As a result, the efficacy of $Li_2O_2$ decomposition is determined to be 3.17$e^-$/$O_2$, which is slightly closer to the ideal 2$e^-$/$O_2$ than that from typical $Li_2O_2$ film from CNT (3.31$e^-$/$O_2$, Supplementary Figure 9)[8,21,24] owing to the lower RC potential. Nevertheless, the deviations from the 2$e^-$/$O_2$ indicates the accompaniment of side reactions as evidenced by the side products of lithium carbonate ($Li_2CO_3$, 532.8 eV) and carboxylate-related bands ($HCO_2Li$, 531.6 eV) emerging in the TEY mode of the O K-edge XANES spectra at 0.5RC and PFY mode at 0.75RC, respectively (Fig. 3a). These side products result from the deteriorating carbon electrode and electrolyte due to instabilities with the superoxide intermediate during DC[7,9] and singlet oxygen produced during RC[35]. Decomposition of the side products requires over ~4 V and

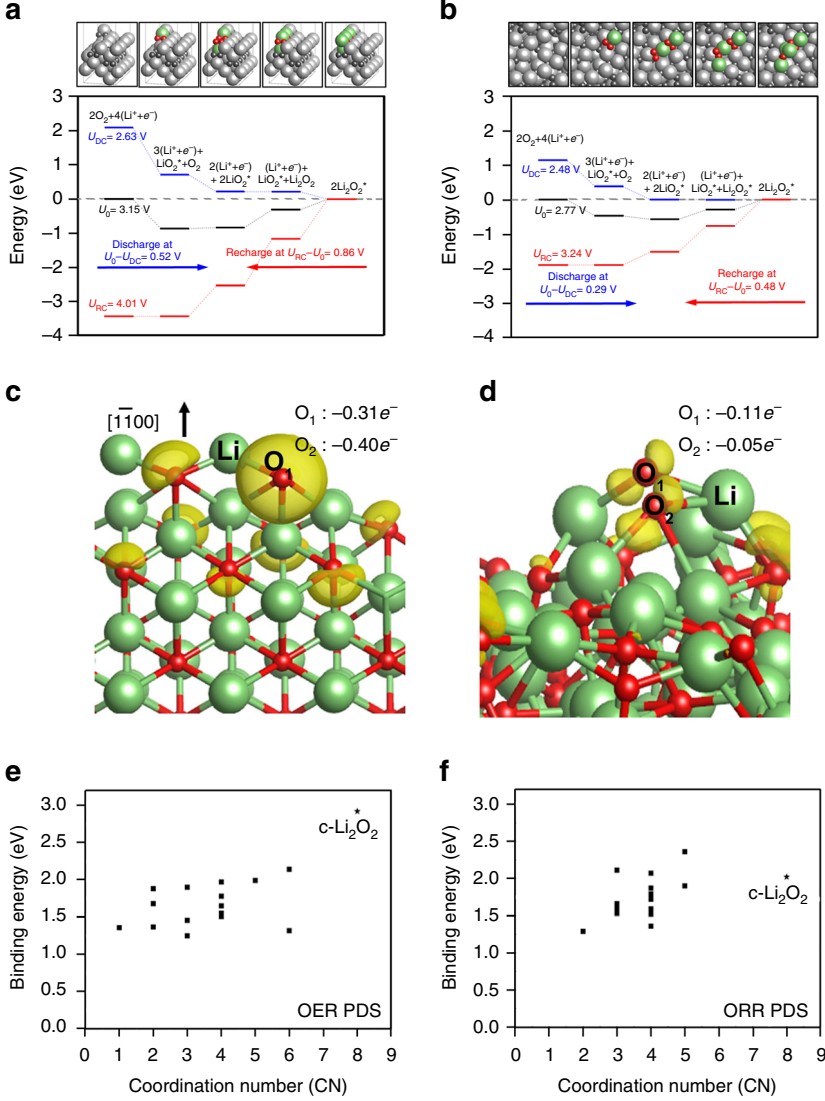

**Fig. 4** Density functional calculations and Bader charge analysis. **a, b** Calculated free energy diagrams **a** on the $Li_2O_2$ ($1\bar{1}00$) and **b** on amorphous $Li_2O_2$, along with the optimized structures. For the amorphous surface, the most favorable adsorption site was considered to construct the diagram. Lithium is colored in light gray (bulk) and light green (adsorbate); oxygen is colored in dark gray (bulk) and red (adsorbate). **c, d** The variation in electron density upon the first $LiO_2$ adsorption on **c** the crystalline $Li_2O_2$ ($1\bar{1}00$) and **d** amorphous $Li_2O_2$ surfaces. The charge density indicated by surface contour is plotted with a threshold value of 0.01e $Å^{-3}$. The newly adsorbing *$LiO_2$ is denoted explicitly as Li, $O_1$, and $O_2$. $O_2$ which is not shown in **c** is behind $O_1$. Lithium is in light green and oxygen in red. **e, f** Binding energies of **e** 1st $LiO_2$ and **f** 2nd $LiO_2$ for 16 sites of the amorphous $Li_2O_2$ surface plotted versus coordination number. Binding energies on the crystalline surface are also shown for comparison, denoted as c-$Li_2O_2$

is reflected in the potential rise at 0.75RC, which directly occurs after the disappearance of $Li_2O_2$ from the CMK-3 surface and is met with $CO_2$ gas evolution[36] (Fig. 3b, d). The residual side products present after RC can compromise cycling performance (Supplementary Figure 10), which affirms the need for more stable electrolytes and electrode materials.

**Morphological and structural effects.** Overall, the 1-D shape with ultrathin amorphous structure plays a pivotal role for facile decomposition. We could confirm this further by modulating the DC current rate to control the product structure. The SEM images in Fig. 5a reveal different $Li_2O_2$ morphological shapes with respect to the DC current rate. As the current rate increases from 10 to 100 mA $g^{-1}_{carbon}$, the flake-like product becomes thinner and less numerous, then finally the conformal film shape is prominent when the current rate is over 100 mA $g^{-1}_{carbon}$. At

higher current rates, there is the swift deposition of $Li_2O_2$ over the surface of CMK-3[37], which nullifies the interior of the electrode by blocking the entry of the mesoporous channels. The XRD results show amorphous characters of both flake and film products (Supplementary Figures 3 and 11), thus their different shapes can be directly correlated to their decomposition behaviors which can be explored with anodic linear sweep voltammetry (LSV) (Fig. 5b). When flake-abundant $Li_2O_2$ is present, a significant anodic peak appears at ~3.18 ± 0.04 V, denoted as $E_{a1}$, while the peak at higher potential of ~3.40 ± 0.03 V, $E_{a2}$, becomes progressively pronounced when increasing proportions of the film are present. Notably, $E_{a2}$ is greater than $E_{a1}$ at over 100 mA $g^{-1}$. The corresponding OEMS analysis demonstrates that the amount of $O_2$ evolution is proportional to the peak intensity of $E_{a1}$ and $E_{a2}$, and decouples this from $CO_2$ evolution at $E_{a3}$ (Fig. 5c). Therefore, it can be concluded that there is prior decomposition of the flake-like products comprised of

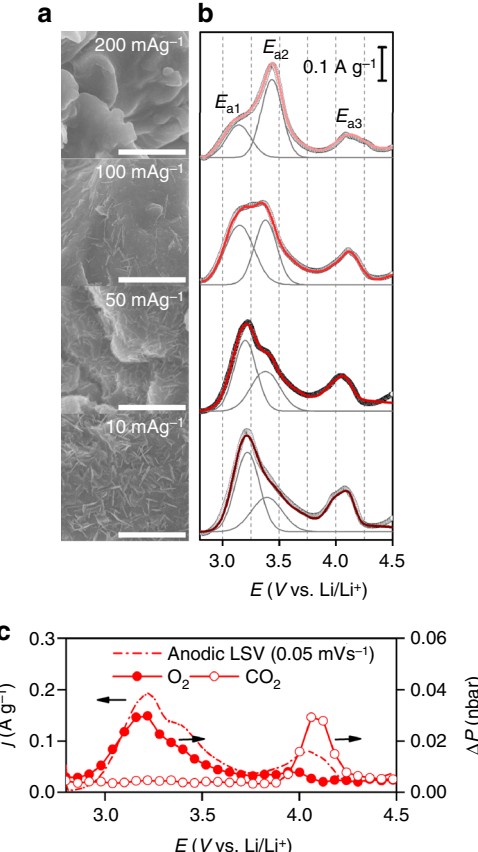

**Fig. 5** Dependence of Li$_2$O$_2$ morphology and corresponding RC profiles on DC current rate. **a** SEM images of CMK-3 electrodes with different current rates of 10, 50, 100, and 200 mA g$^{-1}$$_{carbon}$ (from bottom to top) at a fixed DC capacity of 500 mAh g$^{-1}$$_{carbon}$. The scale bars indicate 1 μm. **b** Corresponding anodic LSV profiles at a sweeping rate of 0.05 mV s$^{-1}$. The black open circles represent experimental LSV curve whereas the red lines indicate the sum of the deconvoluted curves. The deconvoluted peaks above 4.0 V have been omitted for simplicity. **c** Representative in situ gas analysis linked with anodic LSV for Li$_2$O$_2$ formed at a DC current rate of 50 mA g$^{-1}$$_{carbon}$

1-D nanostructured Li$_2$O$_2$ as opposed to the conformal films. As for another control experiment, we induced the partial blockage of the mesoporous channels by annealing CMK-3 at 2000 °C in argon (Ar). This high-temperature annealing brings about disordered and clogged mesoporous channels in some parts of CMK-3 and results in the decrease of the total surface area and volume of the electrode by half, while the turbostratic structure and pore diameter are still preserved (Supplementary Figure 12 and Supplementary Table 1)[25]. After DC, crystalline and lump-shaped Li$_2$O$_2$ is observed together with flake-like Li$_2$O$_2$ (Supplementary Figure 13), which may deposit on the clogged surface of the electrode. In the subsequent RC, two potential plateaus are observed at ~3.5 and ~4.3 V with the corresponding O$_2$ evolution (Supplementary Figure 14). The distinctly separate plateaus with large potential gap account for the far slower decomposition of the crystalline lumps, which confirms again the critical role of morphology and structure in promoting the facile decomposition of Li$_2$O$_2$.

**Mechanism of fast RC**. The question is then on the mechanism accounting for the facile decomposition. As oxygen evolution reaction (OER) is the reverse process of ORR, the same DFT free energy profile in Fig. 4a, b can be used to understand the

enhanced OER mechanism from the amorphous structure. The PDS for OER on both crystalline and amorphous surfaces is the reverse of Eq. (1), LiO$_2$$^*$ ⇆ Li$^+$ + $e^-$ + O$_2$$^*$. The $\eta_{OER}$ for crystalline Li$_2$O$_2$ is calculated to be 0.86 V, which is comparable with the experimental values (~0.9 V, Fig. 1). In contrast, the $\eta_{OER}$ for amorphous Li$_2$O$_2$ at the most favorable binding site is calculated to be 0.48 V, almost 50% reduction in overpotential compared to the crystalline case and is consistent with our experiments. With the PDS of OER being LiO$_2$$^*$ ⇆ Li$^+$ + $e^-$ + O$_2$$^*$ for both crystalline and amorphous surfaces, the underlying origin of reduced $\eta_{OER}$ is again the weaker binding of *LiO$_2$ on the amorphous surface, as in ORR, which can then be explained with the smaller number of new coordination that the adsorbed *LiO$_2$ creates with the amorphous surface (Fig. 4c). Remarkably, the effect of amorphous structure on the PDS of OER is much more prominent than that of ORR, explaining the great enhancement of the Li$_2$O$_2$ decomposition on the amorphous phase. Bader Population analysis on the crystalline versus amorphous surfaces before and after *LiO$_2$ adsorption demonstrates that the amount of charge transferred from the surface to the adsorbed *LiO$_2$ is indeed much more moderate on the amorphous surface due to disordered surface geometries and associated weaker electronic interactions with the adsorbates (Fig. 4e, f).

Along with the thermodynamic understanding of the structural effect, the morphological benefit of 1-D and ultrathin nanostructure is significant. Considering that the 1-D Li$_2$O$_2$ is continuous from the interior of mesoporous channel to exterior, the interior would be embedded in the conducting electrode while the exterior part is enclosed with the electrolyte solution. One can expect facile charge transport across Li$_2$O$_2$ that is confined in the mesoporous channels. However, Li$^+$ dissolution which is responsible for the electron transfer (see Eqs. (1) and (2)) is less plausible within the closed mesoporous channels due to the violation of charge balance. Therefore, preferential depletion takes place from Li$_2$O$_2$ that is surrounded by the electrolyte solution (Fig. 7a). This is in good agreement with the O K-edge XANES results showing the disappearance of Li$_2$O$_2$ from the exterior of CMK-3 surface (Fig. 3a). The cyclic voltammogram (CV) analyses of both CMK-3 and non-mesoporous CNT electrode also confirm this hypothesis. Figure 6 and Supplementary Figure 15 show significant cathodic responses for both CMK-3 and CNT during ORR in comparison with the featureless response under Ar environment. The subsequent OER reveals the pronounced anodic response exclusive to CMK-3 in the potential range of 3.2–4.2 V (Fig. 6a). When the total charges of ORR versus OER, i.e., ($Q_{ORR}/Q_{OER}$), are estimated, they are 1.39 and 1.01 for CNT and CMK-3 electrodes, respectively. The charge ratio at unity indicates more facile decomposition of 1-D Li$_2$O$_2$ and also the negligible loss of Li$_2$O$_2$ during OER. This also implies the less favorable depletion of Li$_2$O$_2$ from the interior of channel, which would otherwise no longer anchor the Li$_2$O$_2$ on exterior of the electrode, leading to the physical loss of Li$_2$O$_2$ and increase in the charge ratio.

## Discussion

We propose that the critical advantage of 1-D nanostructured Li$_2$O$_2$ on the exterior of the channel is the extensive accessibility of Li$^+$ in all directions to the electrolyte solution (Fig. 7a), unlike the case of bulk Li$_2$O$_2$ where electrolyte accessibility is restricted to the outer top surface[38,39] (Fig. 7b). Such large surface area is activated by being in equilibrium with the electrolyte solution (see dark blue area in Fig. 7) and reduces the non-active volume of Li$_2$O$_2$. One of the vital steps of decomposition involve electron and charge (hole, $h^+$, in Li$_2$O$_2$) transport giving Li$^+$ dissolution and O$_2$ evolution, which is determined by the

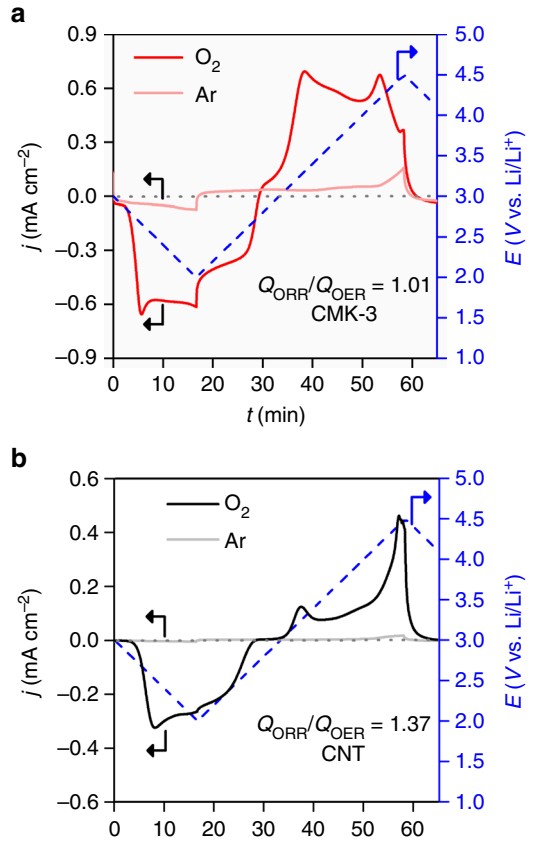

**Fig. 6** Cyclic voltammetry. **a** CMK-3 and **b** CNT electrodes at a sweeping rate of 1 mV s$^{-1}$ in bulk electrolysis cell with 0.5 M LiTFSI/tetraglyme. The blue dashed lines indicates input potential curve. The light and dark solid curves indicate Ar (O$_2$-free) and O$_2$ environment, respectively

corresponding Li$_2$O$_2$ characteristics. Given the 1-D shape of Li$_2$O$_2$, electron transport occurs at highly conductive sites are expected at the interface where Li$_2$O$_2$, electrode, and electrolyte solution are closely in contact (denoted with the star mark in Fig. 7a). From these conductive sites, the charge carriers of holes may swiftly migrate along to the sidewalls of the Li$_2$O$_2$ where highly mobile Li$^+$ and charge carriers lead to facile decomposition, which can account for the decreasing height of nanostructures during RC (Fig. 2f–i). In all, the decomposition rate and behavior of 1-D nanostructured Li$_2$O$_2$ indicate its higher conductivity than bulk Li$_2$O$_2$[40,41]. Electrochemical impedance spectroscopy (EIS) analysis in Supplementary Figure 16 also demonstrates the lowest resistance of 1-D amorphous Li$_2$O$_2$, ~170 Ω for the sum of interface and charge transport resistance, compared with typical amorphous films (~220 Ω) and crystalline Li$_2$O$_2$ lumps (~330 Ω).

The conductive Li$_2$O$_2$ should exhibit high rate capability, which could be examined with increasing RC current rates. Figure 8a reveals that the low RC potential is still maintained below 4 V by 0.75RC from CMK-3 electrode even at 2 A g$^{-1}$. In sharp contrast, the Li$_2$O$_2$ film on CNT leads to the significant rise in potential, reaching ~4.5 V at 0.2RC and 2 A g$^{-1}$ (Fig. 8b). With increasing current rate, the decomposition rate of 1-D Li$_2$O$_2$ is notably distinct from bulk film, implying the exceedingly rapid Li$^+$ dissolution and charge transport. In summary, we have demonstrated the high round-trip efficiency of ~80% in Li–O$_2$ batteries by forming 1-D nanostructures of amorphous Li$_2$O$_2$. This unique shape of Li$_2$O$_2$ could be formed from the guided

growth of the mesoporous carbon channels and retains the ability to suppress RC overpotential even with very high current rates, which is distinct from bulk Li$_2$O$_2$. The initial DC potential profiles correlated with DFT calculations demonstrate that amorphous Li$_2$O$_2$ has lower overpotential for both oxygen reduction and evolution reactions than the crystalline Li$_2$O$_2$ due to disordered geometric arrangements of the surface atoms and associated weaker electronic interactions of the key reaction intermediates, namely lithium superoxide. The 1-D nanostructured shape also has significant benefit for facile decomposition from the highly abundant and facile mobility of Li$^+$ and charge carriers which are present along the surface of Li$_2$O$_2$. This expectation is confirmed from the decreasing height of 1-D structure during the decomposition process and the lowest resistance among other film-shape and bulk Li$_2$O$_2$. This study shows an alternative strategy to surmount the sluggish decomposition of Li$_2$O$_2$ by controlling its shape and structure, and paves the way to promote the facile decomposition of Li$_2$O$_2$ without catalysts.

## Methods

**Synthesis of CMK-3**. CMK-3 was synthesized with reference to the procedure reported by Terasaki and co-workers, but slightly modified[42]. The SBA-15 silica was used as the template, and the carbon source of phenol–formaldehyde mixture was employed to impregnate the silica template.

The SBA-15 template was prepared by using tetraethyl orthosilicate (TEOS) and triblock co-polymer poly(ethylene oxide)–poly(propylene oxide)–poly(ethylene oxide) (PEO–PPO–PEO, Pluronic® P123, Sigma Aldrich, $M_n$ ~5800)[43]. Briefly, 1.0 g of triblock co-polymer surfactant was dissolved in 30 mL of 2.0 M hydrochloric acid (HCl, Wako Pure Chemicals, 35–37%) and after full dissolution 2.2 g of TEOS (Wako Pure Chemicals, ~95%) was added. The mixture was stirred at 40 °C for 24 h and transferred to a Teflon-lined autoclave for hydrothermal reaction at 100 °C for 24 h. The resulting solid precipitate of SBA-15 was filtrated, thoroughly washed with de-ionized (DI) water, and dried at room temperature. Afterward, calcination was carried out at 600 °C in air for the complete removal of organic residues. The SBA-15 was then impregnated with carbon precursors; 0.5 g of as-prepared SBA-15 was added to aqueous solution (6.0 mL DI water) including 0.6 g of phenol (Wako Pure Chemicals, 99%) and 0.5 g of formaldehyde (Wako Pure Chemicals, 36–38%). After stirring at room temperature for 30 min, 100 μL of neat H$_2$SO$_4$ (Wako Pure Chemicals, 98%) was added and the temperature was increased to 80 °C. The vigorous stirring continued until the mixture became dry and the dry precipitate was heated at 150 °C for 3 h. These steps were repeated one more time but with different amount of carbon precursors: 0.3 g of phenol and 0.25 g of formaldehyde. The carbonization was conducted by pyrolysis at 950 °C for 6 h in Ar. Lastly, SBA-15 template was completely etched by soaking in 10% aqueous HF (hydrofluoric acid, Wako Pure Chemicals, 46–48%) for 10 h with stirring. This etching process was carried out for two times. The silica-free mesoporous carbon, i.e., CMK-3, was collected by centrifugation after washing with DI water for several times and dried at 80 °C under vacuum.

**Synthesis of LPC**. A macroporous carbon, LPC (pore diameter ($d$) ~80 nm) was synthesized by using a hard template of silica NP array that was synthesized with reference to Stöber process[44]. Briefly, a mixture of 4.0 mL DI water and 4.0 mL of 25% (v/v) aqueous NH$_3$ (Wako Pure Chemicals) was slowly added to the TEOS solution comprised of 6.0 g of TEOS in 100.0 mL ethanol. The mixture was stirred in a sealed bottle for 20 h at room temperature, showing that the transparent solution became turbid due to formation of silica NPs. After drying the solvent at 70 °C, the collected silica NPs (diameter ($d$) ~80 nm) were sintered at 800 °C for 20 min under air, during which an array of silica NPs was formed[45]. The following processes for impregnation of carbon source, carbonization, and removal of silica NP template were the same as the one described for synthesis of CMK-3.

**Synthesis of Co$_3$O$_4$ NPs on CNT (Co$_3$O$_4$/CNT)**. Co$_3$O$_4$ NPs ($d$ = 8–10 nm) were synthesized according to the procedure reported by Yamada et al.[46] but slightly modified; 0.3 g of cobalt acetate tetrahydrate (Co(C$_2$H$_5$O$_2$)$_2$·4H$_2$O, Wako Pure Chemicals, 99%) was dissolved in 30 mL DI water and 2 mL of 25% (v/v) aqueous NH$_3$ solution was added. After 20 min stirring at room temperature, 100 μL of H$_2$O$_2$ (Wako Pure Chemicals, 30–35.5%) solution was added. By continuing stirring for another 12 h, Co$_3$O$_4$ NPs were formed, which were collected by centrifugation followed by drying at 80 °C in air. To prepare the Co$_3$O$_4$/CNT electrodes, the Co$_3$O$_4$ NPs were mixed with multi-walled CNT (MWCNT, Sigma Aldrich, outer diameter versus length is 7–15 nm versus 0.5–10 μm) with ~40 wt% by grinding using mortar and pestle.

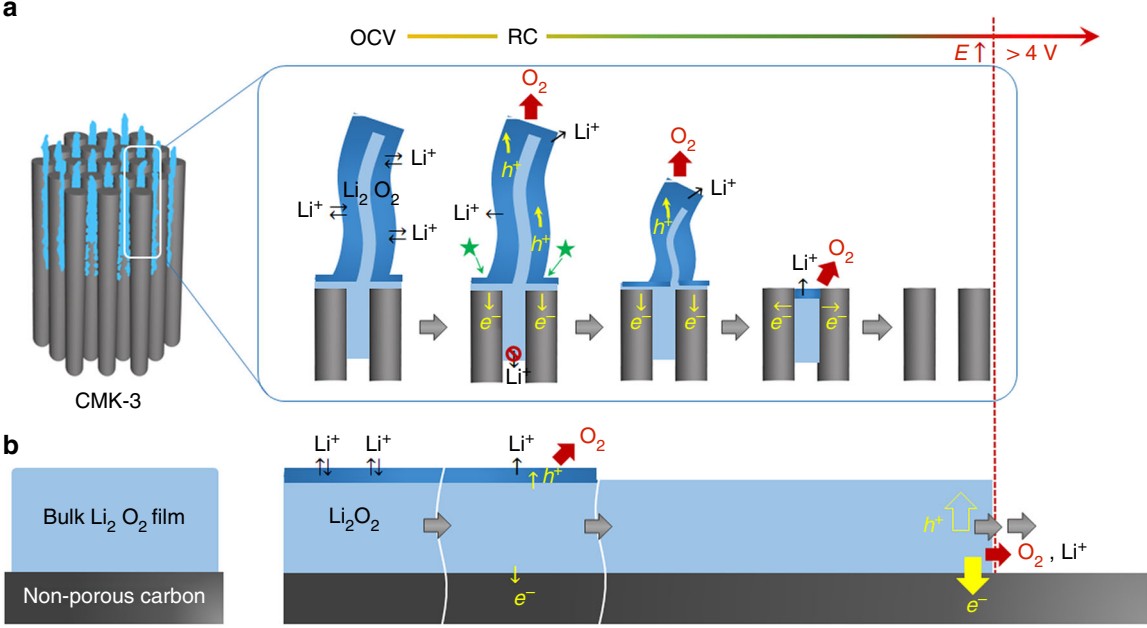

**Fig. 7** Schematic illustration for decomposition processes. **a** 1-D nanostructured and **b** bulk $Li_2O_2$ film. The dark blue region that is distinguished from inside of $Li_2O_2$ with the light blue color indicates activated $Li_2O_2$ surface where free access of $Li^+$ is allowed. The star mark denotes highly conductive sites where $Li_2O_2$, electrode, and electrolyte solution closely contact. Once RC commences, facile $Li^+$ dissolution and charge transport takes place along the activated $Li_2O_2$ surface, which results in $O_2$ evolution and decomposition of $Li_2O_2$ at low potential. The charge carrier hole in $Li_2O_2$ and electron are denoted as $h^+$ and $e^-$, respectively. Since $Li^+$ dissolution of $Li_2O_2$ enclosed within the mesoporous channel would violate charge balance rule, this part of $Li_2O_2$ decomposes later. The bulk $Li_2O_2$ film decomposes sluggishly due to the limited surface area of $Li_2O_2$.

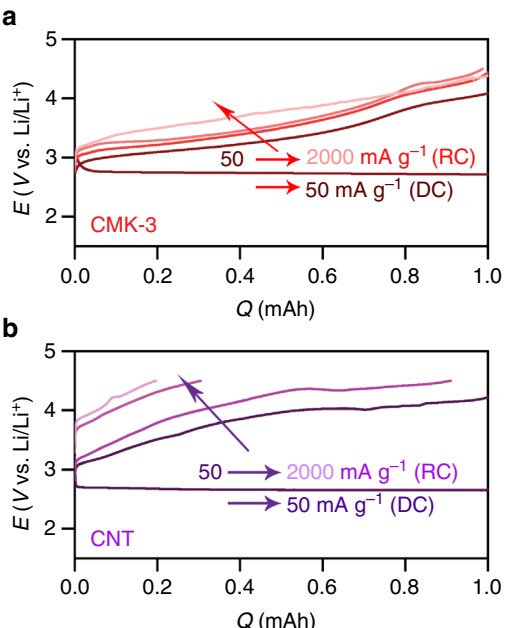

**Fig. 8** Galvanostatic rate capability. **a** CMK-3 and **b** CNT electrodes after DC at 50 mA g$^{-1}$$_{carbon}$ and capacity of 1.0 mAh. The RC was performed at a current rate of 50, 500, 1000, and 2000 mA g$^{-1}$$_{carbon}$ (from dark to light color)

**Synthesis of Ru/CNT.** To synthesize Ru NPs, 0.3 g of $RuCl_3\cdot nH_2O$ (Wako Pure Chemicals) was dissolved in 20 mL ethylene glycol (EG) and the pH was adjusted to 11 by adding 0.1 M NaOH. After stirring for 30 min, temperature was increased to 200 °C and solvothermal reaction was carried out for 4 h in $N_2$ atmosphere. The Ru NPs were incorporated onto MWCNT by dispersing 45 mg of MWCNT in 80 mL ethanol and adding as-prepared Ru colloidal solution to be ~40 wt% loading. A 0.1 M HCl was added to adjust pH to ~5 and the solution was kept under stirring

for 6 h. Subsequently, Ru/CNT was centrifuged, washed with DI water–ethanol mixture (1:1 v/v) for several times, and finally dried at 80 °C under vacuum.

**Fabrication of various electrodes.** Slurries of the carbon materials CMK-3 and LPC were prepared by mixing with LITHion™ (Ion Power) binder (4:1 ratio (w/ w) for carbon/LITHion) in N-methyl-2-pyrrolidone (NMP). They were tape casted to porous carbon paper (Toray, TGP-H-030) using doctor blade, dried slowly at room temperature followed by at 80 °C under vacuum, and finally punched to disks (d = 12 mm). KB (EC-600JD) carbon was mixed with LITHion™(Ion Power, 4:1 ratio (w/w) for carbon/LITHion) in isopropanol (IPA) by 10 min sonication. The resulting slurry was sprayed on glass fiber (GF/C, Whatman) and dried at room temperature. This carbon-coated glass fiber was punched to disks (d = 12 mm). Free-standing CNT electrodes were prepared by dispersing MWCNT powder in IPA followed by tip sonication for 10 min. The $Co_3O_4$/CNT and Ru/CNT were also prepared as a slurry by dispersing in IPA using tip sonication. The slurry for CNT, $Co_3O_4$/CNT, and Ru/CNT was filtered through glass fiber (GF/C) under vacuum, dried at room temperature, punched to disks (d = 12 mm), and peeled off from the glass fiber. All electrodes were completely dried at 200 °C for 5 h using a Büchi oven equipped with vacuum and transferred to Ar-filled glovebox without any exposure to air. The carbon loading mass for all electrodes was 1.0 ± 0.2 mg.

**Assembly of Li–O$_2$ cells and electrochemical measurements.** All assembly processes were carried out in Ar-filled glove box ($O_2$, $H_2O$ < 1 ppm). The cell components were dried at 80 °C under vacuum for 8 h. The Li–$O_2$ cell was assembled with the negative electrode of metallic Li (Honjo) rolled on a stainless steel current collector (d = 12 mm), two separators of Celgard® 2500 (thickness (t) = 25 μm, d = 26 mm) and GF/C (d = 16 mm), and the positive electrode of as-prepared carbon samples. The non-aqueous electrolyte solution of 0.5 M LiTFSI (Lithium bis(trifluoromethanesulfonyl)imide, Kanto Chemical, >99.7%) in tetra-ethylene glycol dimethyl ether (tetraglyme, battery grade, UBE) was added with each 70 μL to three interfaces of Li|Celgard|GF|C|carbon electrode. The as-received tetraglyme was dried using molecular sieves to reduce $H_2O$ content <10 ppm (measured from Karl Fischer titration) before cell assembly. The assembled cell was taken out from the glove box and filled with $O_2$ gas (>99.99995% Tomoe Shokai). The Li–$O_2$ cells were examined with galvanostatic electrochemical testers (WonATech WBCS3000L) in a constant-temperature incubator (25 °C).

**Characterizations.** All post-mortem electrode samples were transferred using hermetic transfer vessel and loaded into characterizing instruments without any exposure to air except microscopy observations. The morphological features of carbon electrodes and DC products were observed using field-emission scanning electron microscope (FE SEM, Hitachi S 4800) and transmission electron

microscopes (TEM) with JEOL JEM-1230 (accelerating voltage 80 kV) and JEOL JEM-ARM200F (accelerating voltages 200 and 80 kV). The loading time for samples was minimized to avoid sample contamination from air. During TEM analysis of the cycled electrodes, the electron beam intensity was carefully adjusted by controlling the aperture size to avoid any damage to the thin $Li_2O_2$ due to incident beam. The $N_2$ adsorption–desorption isotherm was obtained from BEL Japan Inc. BELSORP-mini II and the surface area was calculated by Brunauer–Emmett–Teller (BET) method, pore size distribution and pore volume were estimated by Barrett–Joyner–Halenda (BJH) method. Powder XRD pattern (PXRD) was collected from RigakuSmartLab X-ray diffractometer. The XANES spectra were acquired at the beamline 11 of the Synchrotron Radiation (SR) center of Ritsumeikan University in Japan. FTIR spectroscopy was also performed for chemical analysis of products by using an FTIR spectrometer (Thermo Fisher Scientific Nicolet iS50) equipped with Ar-filled glovebox. For in situ gas analysis, OEMS was used during RC and pressure change was recorded during DC. This custom-built gas analysis system was programmed to automatically detect the collecting gas evolving during RC and the pressure change was concurrently measured for quantitative gas analysis. The detailed description of the OEMS setup can be found in our previous report[24].

**Calculation details**. Spin polarized DFT calculations were performed using the Vienna ab initio simulation package (VASP)[47,48] code with the revised Perdew–Burke–Ernzerhof (RPBE)[49] exchange-correlation functional. The potentials of the atoms were described by projector-augmented wave (PAW)[50]. Throughout this study, we used a cut-off energy of 400 eV. The DFT calculations included two cycles of $Li_2O_2$ formation since the calculation unit cell contains two formula units of $Li_2O_2$ product. The free energy of $O_2$ was indirectly calculated according to the water-splitting reaction, and the energy of $(Li^+ + e^-)$ is assumed to be in equilibrium with the bulk Li, in a way similar to the Computational Hydrogen Electrode method[51]. For the crystalline structure, the reconstructed $Li_2O_2$ $(1\bar{1}00)$ surface reported was used[30]. The Brillouin zone was sampled with a $2 \times 2 \times 1$ Monkhorst-Pack mesh. The bottom two layers were fixed to their bulk positions, with the top two layers fully relaxed. For the amorphous $Li_2O_2$ model, we adopted the amorphous structure predicted by Tian et al.,[23] which described precisely the Li–$O_2$ DC product in recent isotopic labeling expreiments. To reduce the computational cost, 144 atoms (72 Li, 72 O) from the top surface out of 256 atoms were used to make a supercell and half of the atoms from the bottom were fixed. To minimize the calculation artifacts on the amorphous case, we systematically scanned all possible adsorption sites with a $4 \times 4$ grid. Gamma $k$-points were used. The free energies of intermediates at zero potential and pH = 0, $\Delta G = \Delta E + \Delta ZPE - T\Delta S$, was evaluated as follows: The reaction energy $\Delta E$ is calculated using DFT. The zero point energy and entropy terms were neglected considering very small changes in those terms along the reaction in the solid phases. The Bader charge analysis was performed to estimate the amount of charge transfer before and after the adsorption of $LiO_2$ in which the charges of isolated $LiO_2$ were used as a reference[52]. For both crystalline and amorphous models, the 15 Å vacuum space was used in the z-direction. The convergence of energy and forces were set to $1 \times 10^{-5}$ eV and 0.05 eV Å$^{-1}$, respectively.

**Data availability**. The data that support the plots within this paper and other findings of this study are available from the corresponding author on request.

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

## Acknowledgements

This work is financially supported by RIKEN, JST ALCA-SPRING, the National Research Foundation (NRF) of Korea (Grant NRF-2016R1C1B2008690, NRF-2016M3D1A1021147, and NRF-2017R1A2B3010176), and the Nano Material Technology Development Program through the NRF funded by the Ministry of Science, ICT and Future Planning (Grant 2009-0082580). A.D. is grateful for financial support from BK 21 plus fellowship through NRF funded by the Ministry of Education of Korea. The synchrotron XANES experiments at the SR center, Ritsumeikan University, were performed with the approval of "project for creation of research platforms and sharing of advanced research infrastructure of MEXT" (Proposal No. R1432). The authors thank Prof. Seong-Ho Yoon from Kyushu University for high-temperature annealing of CMK-3 and National Institute for Materials Science (NIMS) Battery Research Platform for assistance with TEM observation.

## Author contributions

A.D. conceived the research and carried out the syntheses, electrochemical tests, and characterizations. R.A.W. performed the OEMS experiments and analyses. W.P. and Y.J. performed and interpreted the DFT calculations. K.Y. and T.O. performed the XANES measurements and analyses. H.R.B. supervised the overall research. All authors discussed the experiments and final manuscript.

## Additional information

**Competing interests:** The authors declare no competing financial interests.

