## [Peer Review File · Nature Communications]

Reviewers' comments:

Reviewer #1 (Remarks to the Author):

This paper reports on the use of a templated mesoporous carbon (CMK-3) to constrain the growth of Li₂O₂ during discharge in Li-O₂ batteries. The hypothesis is that amorphous, 1D Li₂O₂ can provide improved charging behaviors (higher Li⁺ conductivity and lower charging overpotentials) compared to crystalline, bulk Li₂O₂. The authors indeed show an impressive charging behavior with very low overpotentials ($E \sim 3$ V for the majority of charge). The authors further studied the growth and decomposition process of these 1D Li₂O₂ nanostructures using a set of techniques (i.e. TEM, SEM, XANES, FTIR and on-line electrochemical mass spectrometry), and concluded that these 1D Li₂O₂ nanostructures are indeed oxidized at lower potential.

These findings compare impressively with other carbon materials and would represent the lowest charging potential reported for Li₂O₂ decomposition. This would be an outstanding new finding so long as the mechanisms are clearly elucidated. However, it appears that CMK-3 materials yield somewhat complex behaviors and therefore are not an ideal model system for studying charge mechanisms. Additionally, the relevance for real Li-O₂ batteries requiring facile transport and good volumetric behavior is not fully clear. Therefore, before publication can be considered, the manuscript needs to clarify several important points and present more compelling evidence or discussion to remove some current ambiguities.

1. The SEM and TEM images are difficult to interpret. The point of view for both techniques with respect to the schematic in Figure 2(i) should be indicated. Also, the 1D nature of the deposits is not evident in the main text TEM images and therefore it is not possible to see how they emerge from or relate to the pores.
2. Similarly, it would help to point out the important features in SEM images. They are currently difficult to relate to the TEM images due to the difference in scale. It would help greatly to include SEM at 0 DC in the main text.
3. In Figure 2i, the authors propose that Li₂O₂ grows from inside the pore channel. However, Figure 4a also indicates that Li₂O₂ can also directly grow from the exterior of the channels at higher current density. Can the authors further explain why at lower current density, initial Li₂O₂ deposition occurs preferentially inside the pore channel where there would be greater transport limitation?
4. Additionally, can the 1D structures also originate from the top surface of the CMK-3? Their diameters (6-15nm) appear larger than the CMK-3 pores. There could be two parallel growth processes occurring, which might have different structures.
5. From the SEM images, it appears that the internal pores are not well-utilized under typical conditions. From Fig. 4a, at the lowest currents of 10 mA/g, at small capacities (500 mAh/g) the Li₂O₂ is visible outside of the pores. However on p. 7, line 157, the authors make a connection between the observed capacity at full discharge and the theoretical capacity, which implies that a high fraction of the internal pores are filled. These seem to be in disagreement.
6. Conventional wisdom would predict that ~ 4 nm pores cannot be well utilized as they would rapidly become blocked by solid Li₂O₂. How do the authors motivate the picture of a continuously growing and emerging 1D structure? Does it imply continued nucleation in the pores? Is electrolyte displacement significant?
7. Is it possible that the small clusters of Li₂O₂ formed in the pores are easily oxidized due to improved electronic contact with the carbon? The authors invoke a delithiation mechanism, but one significant difference between their electrode materials and other particle-based or CNT-based materials is the well-defined and ordered internal carbon surface area. Does the fraction of low charging voltage scale with how much Li₂O₂ formed within the pores (or the inverse, how much formed outside?). This could provide a support for a mechanism.

8. Overall, it is suggested to present all of the structural information about CMK-3 upfront as it currently appears disjointed in the text and follows behind the electrochemical data.
9. CMK-3 was also used by Part et al. (ChemSusChem 2015, 8, 3146), but such a low charge voltage was not reported. Could the authors explain what makes their CMK-3 material different from previously reported ones so that such smaller charge voltages can be achieved?
10. Page 4 Line 76-79: the authors need to specify more clearly whether the current is normalized to the gram of carbon material or total cathode mass since several cathodes also contain substantial amounts of binders and catalyts.
11. In comparing charging potentials between CMK-3 and CNTs, are differences in thickness of the Li₂O₂ important? If the "1D" structure of Li₂O₂ is proposed to be significant with CMK-3, the authors should present comparable data for CNT electrodes and comment on the degree of amorphization and side products.
12. In the text, the XANES data are currently interpreted to indicate that "the nanostructured Li₂O₂ over the exterior of CMK-3 surface is decomposed prior to the Li₂O₂ inside the channels" (p. 7, line 136) because the TEY signal changes more significantly (earlier in charge) than the FY. Some discussion on escape depth and volume being probed in XANES measurements should be included to support this interpretation. Additionally, the TEY data seem to strongly indicate that what was initially Li₂O₂ on the surface of deposits is converted to formate or carbonate during charge, but this is not discussed.
13. There seems to be evidence of parasitic reactions even at low charging overpotentials. In Figure 3d, why is the O₂ evolution rate always lower than expected at 2 e⁻/O₂? The authors should compare with other works.
14. Figure S6 seems to indicate significant amounts of formate and acetate are formed during discharge, however this is not given adequate attention in the text. What is the role of parasitic products and how does it affect the interpretation of results?
15. P. 10, line 222, how is continuous O₂ evolution indicative of a Li⁺ dissolution process? The writing here was difficult to understand.
16. What is the theoretical volumetric capacity of these materials and can they be considered promising for Li-O₂ battery cathodes? What is the relevance to advance the field closer to yielding a real Li-O₂ battery?
17. P. 9, lines 196-198, "This clearly verifies O₂ ingress into the mesoporous channels of CMK-3 ... as shown by higher capacitance in CV under O₂-free atmosphere." This statement is difficult to understand, please elucidate.
18. P. 9, lines 208-209, how does the total Q_{ORR}/Q_{OER} ratio indicate that surface Li₂O₂ charges first?

Reviewer #2 (Remarks to the Author):

In this work, the authors show the design of lithium peroxide to one-dimensional and amorphous nanostructure using the mesoporous carbon electrode, and its pronounced effects for lithium-oxygen battery performance. This work also shows unprecedented high round-trip efficiency and fast charging for mesoporous carbon cells, which verify the rapid decomposition of nanostructured lithium peroxide during charging. Moreover, it is inferred that the enhanced battery round-trip efficiency should be attributed to the formation of amorphous Li₂O₂, which further increases the ionic conductivity. Strong supporting data from snapshots of microscopic images, various cycling voltammetry and battery tests, and on-line electrochemical mass spectrometry confirmed the formation and the role of nanostructured lithium peroxide for fast decomposition. Overall this work is interesting and the detailed characterizations could shed light on the chemistry of Li-O₂ battery in different systems. I recommend its publication after addressing the following comments.

1. In Figure 1, why are the discharge profiles almost the same using different carbon electrodes? Now that the charge process is greatly dependent on the carbon electrodes, why is the discharge process not affected?
2. It is appealing to achieve enhanced round-trip efficiency thanks to the formation of amorphous 1-D Li₂O₂. Would the amorphous Li₂O₂ still be generated at high discharge rate? If so, how to improve it? Relevant data or insightful discussion need be added.
3. The authors claimed that amorphous Li₂O₂ give rise to increased ionic conductivity. This point can be made more convincing with more evidences by EIS test or other characterization. And the authors are suggested to provide more systematic cycling performance.

Reviewer #1 (Remarks to the Author):

This paper reports on the use of a templated mesoporous carbon (CMK-3) to constrain the growth of Li_2O_2 during discharge in Li-O₂ batteries. The hypothesis is that amorphous, 1D Li_2O_2 can provide improved charging behaviors (higher Li^+ conductivity and lower charging overpotentials) compared to crystalline, bulk Li_2O_2 . The authors indeed show an impressive charging behavior with very low overpotentials ($E \sim 3$ V for the majority of charge). The authors further studied the growth and decomposition process of these 1D Li_2O_2 nanostructures using a set of techniques (i.e. TEM, SEM, XANES, FTIR and on-line electrochemical mass spectrometry), and concluded that these 1D Li_2O_2 nanostructures are indeed oxidized at lower potential.

These findings compare impressively with other carbon materials and would represent the lowest charging potential reported for Li_2O_2 decomposition. This would be an outstanding new finding so long as the mechanisms are clearly elucidated. However, it appears that CMK-3 materials yield somewhat complex behaviors and therefore are not an ideal model system for studying charge mechanisms. Additionally, the relevance for real Li-O₂ batteries requiring facile transport and good volumetric behavior is not fully clear. Therefore, before publication can be considered, the manuscript needs to clarify several important points and present more compelling evidence or discussion to remove some current ambiguities.

Response: We thank the reviewer for the insightful comments, which gave us a new point of view. According to reviewer's suggestion, firstly we have significantly modified the overall manuscript to clarify the Li_2O_2 formation and decomposition process. In the revised manuscript, we also have added computational simulations to elucidate the thermodynamics of key reactions during DC and RC. Here we address the reviewer's comments point by point.

Comments 1-2: The SEM and TEM images are difficult to interpret. The point of view for both techniques with respect to the schematic in Figure 2(i) should be indicated. Also, the 1D nature of the deposits is not evident in the main text TEM images and therefore it is not possible to see how they emerge from or relate to the pores.

Similarly, it would help to point out the important features in SEM images. They are currently difficult to relate to the TEM images due to the difference in scale. It would help greatly to include SEM at 0 DC in the main text.

Responses 1-2: We agree with the reviewer's comment that there is a gap between the SEM and TEM images due to their different resolutions. TEM images show ultrathin Li_2O_2 consisting of one-dimensional nanostructures, while SEM images display a contour of their entangled form. To clarify these two different features and avoid confusion, we have mainly showcased the TEM images in Figure 2 and separated the SEM images to Figure S4 in the revised manuscript. The previous TEM image for deposition of product at initial DC has been replaced with more apparent 1-D product image in Figure 2b in the revised manuscript. In addition, we have added SEM image of 0DC CMK-3 to Figure S4 with other SEM images. Although the 1-D nanostructures of Li_2O_2 are clear in Figure 2b–d, it is difficult to distinguish precisely where the Li_2O_2 emerges, because in the TEM images, the contrast of the CMK-3 surface becomes darker with the formation of products. Instead we have suggested other supporting

evidence on the mesoporous channel-guided growth of nanostructured Li_2O_2 , which will be discussed in responses 3 and 4.

[Page 5] Higher-resolution imaging tool of transmission electron microscopy (TEM) address the details of product morphology to ultrathin and 1-D nanostructures that have the width of 6–15 nm (Figure 2a–b). They grow up by loosely entangling with one another during DC (from 0.25DC to 1.0 DC, Figure 2c–e) and their contours display flake-like shape at 1.0 DC (Figure 2e).

[Page 24, Figure 2] **Figure 2.** Transmission electron microscopy (TEM) images of (a) as-prepared, (b–e) discharging and (f–i) recharging CMK-3 surfaces. The scale bars are 50 nm. The top label of image denotes the depth of DC or RC, denoted by Q/Q_{total} at a fixed Q_{total} of 1.0 mAh and the current rate of $50 \text{ mA g}^{-1}_{\text{carbon}}$. The yellow dashed lines indicate the surface of CMK-3. (j) Schematic illustration of initial growth of Li_2O_2 , blue products, from CMK-3 electrode. Over 0.5DC, one-dimensional structure and ultrathin Li_2O_2 grow upwards by loosely entangling with one another, which display flake-like shape (see SEM images in Figure S4).

[Page 5 in SI, Figure S4] **Figure S4.** SEM images of CMK-3 electrodes of (a) as prepared, (b–e) different depth of DC and (f–i) different state of RC denoted by Q/Q_{total} at a fixed Q_{total} of 1.0 mAh and the current rate of $50 \text{ mA g}^{-1}_{\text{carbon}}$.

Comment 3: In Figure 2i, the authors propose that Li_2O_2 grows from inside the pore channel. However, Figure 4a also indicates that Li_2O_2 can also directly grow from the exterior of the channels at higher current density. Can the authors further explain why at lower current density, initial Li_2O_2 deposition occurs preferentially inside the pore channel where there would be greater transport limitation?

Response 3: We believe that Li_2O_2 is formed both interior of the mesoporous channels and the surface of electrode during DC, while the prominent growth process is dependent on the differences in current rate. At low current rate, ultrathin and 1-D Li_2O_2 is predominantly observed (Figure 2). Since this morphology is not found from other non-mesoporous carbon electrodes at the same current rate (Figure S2), we ascribe these the ultrathin and flake-shape Li_2O_2 to the mesoporous channel-guided growth of Li_2O_2 . The driving force for the favorable Li_2O_2 growth emerging from mesoporous channels is presumably due to capillary force: the fast filling of organic adsorbate from micropore to mesopore by capillary force has been previously reported (*Nat. Mater.* **2003**, 2, 473–476). However, when the current rate is increased, the electric field significantly affects the growth process and the resulting rapid nucleation and growth produces conformal film covering the entry of the mesoporous channels. Figure 5a demonstrates the shape transition of Li_2O_2 from ultrathin flake (i.e., entangled 1-D nanostructures) to conformal film with increasing current rate. In the revised manuscript, we have rephrased these different growth processes with respect to the current rate.

[Page 6] With CMK-3, the O_2 gas and electrolyte solution can diffuses into the mesoporous channels by capillary force^{28,29}.

[Page 9] At higher current rates, there is the swift deposition of Li_2O_2 over the surface of CMK-3³⁷ which nullifies the interior of the electrode by blocking the entry of the mesoporous channels.

Comment 4: Additionally, can the 1D structures also originate from the top surface of the CMK-3? Their diameters (6-15nm) appear larger than the CMK-3 pores. There could be two parallel growth processes occurring, which might have different structures.

Response 4: Since the 1-D Li_2O_2 structures could not be found in the non-mesoporous carbon electrodes (Figure S2) and the shape of Li_2O_2 mimics that of the mesoporous channels, the unique morphology of Li_2O_2 highly likely originates from the mesoporous channels in CMK-3.

The diameter of Li_2O_2 is limited from that of mesoporous channel inside CMK-3. However, once it departs from the channel, Li_2O_2 produced from the CMK-3 surface can bind with 1-D Li_2O_2 towards both vertical and horizontal directions. The 1-D nanostructure demonstrates the prominent growth of Li_2O_2 towards the vertical direction, while larger diameter than that of mesoporous channel indicates a certain contribution of lateral growth, which we have rephrased in the revised manuscript.

[Page 7] As the DC proceeds at low DC current rate, amorphous Li_2O_2 grows and emerges from the mesoporous channel. The continuous 1-D growth is also observed on the electrode surface as shown in **Figure 2a–e** and **Figure 2j**, and is similar in shape to the mesoporous channels. Although the prominent vertical growth is apparent, Li_2O_2 species may also bind at the lateral sides of the 1-D Li_2O_2 protruding from the exterior of the mesoporous channels. This can account for the slightly increased width of the nanostructure than the diameter of mesoporous channel.

Comment 5: From the SEM images, it appears that the internal pores are not well-utilized under typical conditions. From Fig. 4a, at the lowest currents of 10 mA/g, at small capacities (500 mAh/g) the Li_2O_2 is visible outside of the pores. However on p. 7, line 157, the authors make a connection between the observed capacity at full discharge and the theoretical capacity, which implies that a high fraction of the internal pores are filled. These seem to be in disagreement.

Response 5: The reviewer's comment of the shallow depth of Li_2O_2 filling in mesoporous channel is right. The appearance of Li_2O_2 at the initial stage indicates fast clogging of mesoporous channel. The comparison of real and theoretical capacities also supports the partial filling of Li_2O_2 because the high capacity measured from the experiments also includes Li_2O_2 on the exterior of mesoporous channel (**Figure S8**). We have rephrased the partial filling of channels in the revised manuscript.

[Pages 7–8] It is noted that the mesoporous channels are only partially filled with Li_2O_2 due to slow O_2 diffusion and rapid clogging by Li_2O_2 ²⁹. This is verified from the full DC capacity of $\sim 2300 \text{ mAh g}^{-1}_{\text{CMK-3}}$ that is lower than the theoretical capacity of $\sim 2430 \text{ mAh g}^{-1}_{\text{CMK-3}}$ estimated from the full volume of the mesoporous channels, despite the inclusion of Li_2O_2 formed over the exterior of the channels (**Figure S8**). Therefore, the initial growth occurring in the vicinity of channel entry is remarkably crucial to lead to the unique shape of Li_2O_2 .

Comment 6: Conventional wisdom would predict that $\sim 4 \text{ nm}$ pores cannot be well utilized as they would rapidly become blocked by solid Li_2O_2 . How do the authors motivate the picture of a continuously growing and emerging 1D structure? Does it imply continued nucleation in the pores? Is electrolyte displacement significant?

Response 6: We agree with reviewer's comment of the occurrence of fast clogging induced by the Li_2O_2 . However, interestingly, 1-D nanostructures of Li_2O_2 also appears on the surface of CMK-3 as shown in Figure 2, which lead us to infer the growth process.

We found the lowest overpotential from CMK-3 among other non-mesoporous electrodes during oxygen reduction reaction (**Figure S6**). The DFT calculation expects weak binding of LiO_2 with low overpotential, on amorphous Li_2O_2 surfaces. In the confined space of mesoporous channel, this LiO_2 can be highly stable, by referring to the report, *ChemCatChem* **2015**, *7*, 738–742, and may build up a chelating network via weak binding interaction (*J. Electrochem. Soc.* **2011**, *158*, A1177–A1184), which may trigger continuous growth along the mesoporous channel. As shown in response 5, Li_2O_2 partially fills the mesoporous channel and notable growth can occur at the vicinity of entry of mesoporous channel.

We do not clearly understand “electrolyte displacement” the reviewer asked. In general, electrolyte displacement occurs from two electrolyte solutions. Since we have used one electrolyte solution and a quantity of H_2O is minor (< 10 ppm of H_2O), there is no possibility of electrolyte displacement. If the reviewers pointed out that the Li_2O_2 displaced the electrolyte in the pore, we could answer its difficulty due to higher density of solid Li_2O_2 that precipitates in the surface of CMK-3 than solvated Li^+ and O_2 gas, and no driving force for 1-D growth.

In the revised manuscript, we have added the DFT result and rephrased the part regarding the growth process.

[Pages 6–7] With CMK-3, the O_2 gas and electrolyte solution diffuses into the mesoporous channels by capillary force^{28,29}, where the confined O_2 is reduced via two-electron transfer as follows:

Here, * represents adsorbed species on the surface. After step 1, Li_2O_2 can form by either step 2 or by the disproportionation of LiO_2 species ($2\text{LiO}_2 \rightleftharpoons \text{Li}_2\text{O}_2^* + \text{O}_2$). At the initial stage of DC with CMK-3, we found that the overpotential of oxygen reduction reaction (ORR), which leads to the formation of amorphous Li_2O_2 in the mesoporous channels, was lower than the crystalline Li_2O_2 formed with the non-mesoporous electrodes (**Figure S6**). The lower ORR overpotential on the amorphous Li_2O_2 surface can be understood by using density functional theory (DFT) calculations on the key intermediates of the electrochemical reactions.

The DFT calculations included two cycles of Li_2O_2 formation since the calculation unit cell contains two formula units of Li_2O_2 product. **Figure 4a** shows the calculated free energy diagram along with the optimized structures at three different applied potentials (U). The free energy of O_2 was indirectly calculated according to the water-splitting reaction, and the energy of ($\text{Li}^+ + e^-$) is assumed to be in equilibrium with the bulk Li, in a way similar to the Computational Hydrogen Electrode method³⁰. The applied potential (U) is incorporated into the energetics via $-neU$, where n is the number of transferred ($\text{Li}^+ + e^-$). The calculated overpotential (η) for ORR is defined as $\eta_{\text{ORR}} = U_0 - U_{\text{DC}}$ where U_0 and U_{DC} are the equilibrium and DC potentials, respectively^{31,32}. Using these prescriptions, the η_{ORR} for the crystalline structure, reconstructed Li_2O_2 ($1\bar{1}00$) that is the most stable surface³¹, is calculated to be 0.52V, which is in good agreement with the experimental value ($\eta_{\text{ORR}} = 0.4 - 0.5$ V, from U_0 of 3.15 V, **Figure 1** and **Figure S6**). For the amorphous Li_2O_2 surface, the structure suggested from Tian et al²³ determined from first-principles molecular dynamics simulations was utilized. We divided the surface into 16 equivalent areas to find the lowest energy adsorption site (**Figure S7**), and the free energy diagram shown in **Figure 4b** is based on the latter most favorable reaction site. However, we note as summarized in **Table S2**, that

statistically 14 sites out of 16 still show lower η_{ORR} compared to that from the crystalline surface, which is in accordance with the experimental results (**Figure S6**).

For both amorphous and crystalline surfaces, the potential determining step (PDS) for ORR is calculated to be step (2), indicating that the weaker binding of second $^*\text{LiO}_2$ adsorption on the amorphous surface is responsible for a lower η_{ORR} . Structurally, this weaker binding of $^*\text{LiO}_2$ on the amorphous surface is due to the disordered arrangements of the surface Li and O atoms, preventing the newly adsorbing $^*\text{LiO}_2$ species from forming optimal coordination with the surface. To quantify this, in **Figure 4f**, we plotted the binding energies of $^*\text{LiO}_2$ for the PDS of ORR as a function of the number of newly formed Li-O bonds as a result of $^*\text{LiO}_2$ adsorption. Overall, the low $^*\text{LiO}_2$ binding energies on the amorphous surface are indeed associated with the low coordination number that the newly adsorbed $^*\text{LiO}_2$ forms with the surface, compared to that on the crystalline surface.

On the basis of the DFT results, we expect the weakly binding LiO_2 to constitute a chelating network within the confined channels^{33,34}, prior to the formation of amorphous Li_2O_2 . As the DC proceeds at low DC current rate, amorphous Li_2O_2 grows and emerges from the mesoporous channel. The continuous 1-D growth is also observed on the electrode surface as shown in **Figure 2a–e** and **Figure 2j**, and is similar in shape to the mesoporous channels.

Figure 4. Density functional calculations and Bader charge analysis. (a–b) Calculated free energy diagrams (a) on the Li_2O_2 ($\bar{1}\bar{1}00$) and (b) on amorphous Li_2O_2 , along with the optimized structures. For the amorphous surface, the most favorable adsorption site was considered to construct the diagram. Lithium is colored in light gray (bulk) and light green (adsorbate); Oxygen is colored in dark gray (bulk) and red (adsorbate). (c–d) The variation in electron density upon the first LiO_2 adsorption (e) on the crystalline ($\bar{1}\bar{1}00$), and (f) amorphous Li_2O_2 surfaces. The charge density isosurfaces contour is plotted with a threshold value of $0.01e \text{ \AA}^{-3}$. The newly adsorbing LiO_2 is denoted explicitly as Li, O_1 , and O_2 . O_2 which is not shown in (c) is behind O_1 . Lithium is in light green and oxygen in red. (e–f) Binding energies of 1st LiO_2 (e) and 2nd LiO_2 (f) for 16 sites of the amorphous Li_2O_2 surface plotted versus coordination number. Binding energies on the crystalline surface are also shown for comparison, denoted as c- Li_2O_2 .

Comment 7-8: Is it possible that the small clusters of Li_2O_2 formed in the pores are easily oxidized due to improved electronic contact with the carbon? The authors invoke a delithiation mechanism, but one significant difference between their electrode materials and other particle-based or CNT-based materials is the well-defined and ordered internal carbon surface area. Does the fraction of low charging voltage scale with how much Li_2O_2 formed within the pores (or the inverse, how much formed outside?). This could provide a support for a mechanism. Overall, it is suggested to present all of the structural information about CMK-3 upfront as it currently appears disjointed in the text and follows behind the electrochemical data.

Responses 7-8: We appreciate the reviewer to bringing up this point. The facile surface decomposition of Li_2O_2 to the Li_2O_2 in the interior of the mesoporous channels has been attested from the following evidences: (1) fast disappearance in the surface of Li_2O_2 observed from TEM images (Figure 2), (2) earlier removal of $\sigma^*(\text{O}-\text{O})$ bands from surface-sensitive TEY mode than that from bulk-sensitive PFY mode (Figure 3a), (3) the unity for the ratio of Q_{ORR} vs Q_{OER} from the CV (Figure 6), indicating negligible physical loss of Li_2O_2 that is improbable if decomposition starts from the inside of mesoporous channels as this would no longer anchor the exterior 1-D nanostructures and (4) violation of charge-balance with Li^+ dissolution into electrolyte solution which is isolated within the mesoporous channel clogged by Li_2O_2 . Although we cannot estimate the precise fraction of Li_2O_2 placed interior and exterior of the CMK-3 due to the technical limitations, we believe that the above-mentioned clear evidences can fully support our proposed mechanism. We have rephrased the mechanism part in the revised manuscript.

[Page 8] We now turn our attention to the RC process. The decreasing height of 1-D nanostructured Li_2O_2 is followed by its complete depletion from CMK-3 surface at 0.75 RC (Figure 2f–i). The O K-edge XANES spectra with surface-sensitive TEY mode are also consistent with this result, showing a significant decrease in $\sigma^*(\text{O}-\text{O})$ band by 0.5RC and disappearance at 0.75RC (light solid line in Figure 3a). However, regarding the interior of CMK-3, the behavior of Li_2O_2 decomposition is different. The bulk-sensitive PFY mode shows the pronounced $\sigma^*(\text{O}-\text{O})$ band at 0.5RC, which decreases at 0.75RC and disappears at 1RC (dark solid line in Figure 3a). These results clearly demonstrate the preferential decomposition of Li_2O_2 exterior of CMK-3, relative to Li_2O_2 in the interior of the mesoporous channels, which we will discuss in more detail below.

[Pages 10–11] Considering that the 1-D Li_2O_2 is connected continuously from the interior of mesoporous channel to exterior, the interior would be embedded in the conducting electrode while the exterior part is enclosed with the electrolyte solution. One can expect fast charge transport across Li_2O_2 that is

confined in mesoporous channel with a small diameter. However, Li^+ dissolution which is responsible for the electron transfer (see step (1) and (2)) is less plausible within the closed mesoporous channels due to the violation of charge balance. Therefore, preferential depletion takes place from Li_2O_2 that is surrounded by electrolyte solution (**Scheme 1a**). This is in good agreement with the O K-edge XANES results showing the disappearance of Li_2O_2 from the exterior of CMK-3 surface (**Figure 3a**).

[Page 10] When the total charge of ORR against OER, i.e., ($Q_{\text{ORR}}/Q_{\text{OER}}$), are estimated, they are 1.39 and 1.01 for CNT and CMK-3 electrodes, respectively. The total charge ratio at unity indicates more facile decomposition of 1-D Li_2O_2 and also the negligible loss of Li_2O_2 during OER. This also indicates the less favorable depletion of Li_2O_2 from interior of channel, which would otherwise no longer anchor the Li_2O_2 on exterior of the electrode leading the physical loss of Li_2O_2 and increase in the charge ratio.

Comment 9: CMK-3 was also used by Part et al. (ChemSusChem 2015, 8, 3146), but such a low charge voltage was not reported. Could the authors explain what makes their CMK-3 material different from previously reported ones so that such smaller charge voltages can be achieved?

Response 9: The difference we could find was electrode components: Park et al. have used CMK-3, Super P, PVDF (6:3:1) whereas we have used CMK-3 and LITHion™ (lithiated nafion) (4:1) without the addition of conductive carbon. It has been reported that PVDF binder is unstable in Li–O₂ cells due to participation of unintended chemical reactions (*J. Am. Chem. Soc.* **2012**, *134*, 2902–2905). The inclusion of Super P could also mask the effect of CMK-3, thus ultrathin and nanostructured Li_2O_2 might be mixed with bulk Li_2O_2 film, which could raise the RC potential. We also point out that Li–O₂ cell performance associated with the shape and structure of Li_2O_2 are quite different from test environments mostly caused by air and H₂O leakage of Li–O₂ cells. Although we do not know the specific test condition from the reference, we have self-confidence in our Li–O₂ system including negligible contaminants, air and H₂O. Moreover, this is the reason why we examined all non-mesoporous carbon electrodes, CMK-3 and catalyst electrodes together at the same conditions rather than referring to other reports' values, which allowed us for fair comparison of cell performance. We have mentioned the use of no carbon additive to the CMK-3 electrode in the revised manuscript.

[Page 4] As the mesoporous carbon electrode in Li–O₂ cells, CMK-3 was employed without the use of any additive carbon.

Comment 10: Page 4 Line 76-79: the authors need to specify more clearly whether the current is normalized to the gram of carbon material or total cathode mass since several cathodes also contain substantial amounts of binders and catalyts.

Response 10: For all tests, the currents were normalized to the mass of carbon material. We have added this description in the revised manuscript.

[Page 23, Figure 1 caption] All electrodes have same carbon mass of ~1.0 mg and galvanostatic tests were performed at a same current rate of 50 mA g⁻¹_{carbon} in 0.5 M LiTFSI in tetraglyme to the fixed capacity of 1.5 mAh.

Comment 11: In comparing charging potentials between CMK-3 and CNTs, are differences in thickness of the Li_2O_2 important? If the “1D” structure of Li_2O_2 is proposed to be significant with CMK-3, the authors should present comparable data for CNT electrodes and comment on the degree of amorphization and side products.

Response 11: The critical factor we have proposed is (1) high surface area of the nanostructured Li_2O_2 , which is possibly has similar meaning of thickness as the reviewer mentioned, and (2) amorphous structure rather than height of Li_2O_2 is generally 20–50 nm for both film and 1-D nanostructures. The ultrathin and nanostructured Li_2O_2 has much higher surface area and lower volume than those of bulk Li_2O_2 film. This implies large contact area enclosed by the electrolyte solution and improvement of Li^+ dissolution throughout all directions. The amorphousness is the other important factor. Figure S3 clearly demonstrates the negligible XRD pattern of Li_2O_2 with CMK-3, while a significant signal appears with CNT, which we mentioned in the main text. Decomposition of both 1-D nanostructure and bulk Li_2O_2 film is accompanied by side products. The side products from CNT electrode were reported from our previous papers (*Nano Lett.* **2016**, *16*, 2969-2974, *Chem. Mater.* **2016**, *28*, 8006-8015). The chemical species of side products are similar in both electrode cases. However, due to the difficulties in quantitative accessing side products from both electrodes and complexity of electrochemical/chemical side reactions producing immiscible solid, miscible solid and gas products, we have decided to defer their detailed comparison in the manuscript.

Comment 12: In the text, the XANES data are currently interpreted to indicate that “the nanostructured Li_2O_2 over the exterior of CMK-3 surface is decomposed prior to the Li_2O_2 inside the channels” (p. 7, line 136) because the TEY signal changes more significantly (earlier in charge) than the FY. Some discussion on escape depth and volume being probed in XANES measurements should be included to support this interpretation. Additionally, the TEY data seem to strongly indicate that what was initially Li_2O_2 on the surface of deposits is converted to formate or carbonate during charge, but this is not discussed.

Response 12: We appreciate the reviewer’s suggestion for detection depth of two different modes and have revised manuscript accordingly.

As the reviewers mentioned, we found side products in the XANES spectra occurring with the depletion of Li_2O_2 during RC. However, there is no evidence for conversion of Li_2O_2 , which is also beyond the scope in our work, which focuses on showing the facile decomposition nanostructured Li_2O_2 with 1-D and amorphous structure.

[Page 5] The O K-edge XANES spectra demonstrate the notable $\sigma^*(\text{O}-\text{O})$ bands at 530.5 eV arising from Li_2O_2 ²⁶ in both total electron yield (TEY) and partial fluorescence yield (PFY) modes (**Figure 3a**), which has the escape depth of tens and hundreds of nanometers, respectively.

Comment 13: There seems to be evidence of parasitic reactions even at low charging overpotentials. In Figure 3d, why is the O_2 evolution rate always lower than expected at 2 e-/O₂? The authors should compare with other works.

Response 13: With CMK-3, the electrons per oxygen molecule (e^-/O_2) obtained from OEMS is $3.17 e^-/O_2$, which is higher than ideal $2 e^-/O_2$ due to inevitable side reactions likely arising from the superoxide intermediate during DC and singlet oxygen during RC (Nat. Energy 2017, 2, 17036). The side products of carboxylates and carbonates suggest degradation of the carbon electrode and electrolyte solution (Chem. Rev. 2014, 114, 11721-11750).

In addition, we have added OEMS result of bulk and crystalline Li_2O_2 film from CNT electrode in the revised manuscript, which is shown to be $3.31 e^-/O_2$ at the similar test condition (Figure S10). The higher value of e^-/O_2 from CNT than that from CMK-3 can be attributed to more unintended reactions, which is closely associated with the higher RC potential. We did not compare our result from other laboratories due to the differences in test conditions (see **response 9**). We note however, that the OER/ORR ratio of ~60% is essentially in line with literature (ACS Appl. Mater. Interfaces, 2017, 9(31) 25976–25984, J. Phys. Chem. Lett., 2011, 2 (10), 1161–1166, ACS Nano, 2014, 8 (12), 12483–12493).

[Pages 8–9] As a result, the efficacy of Li_2O_2 decomposition is determined to be $3.17 e^-/O_2$, which is closer to the ideal $2 e^-/O_2$ than that from typical Li_2O_2 film from CNT ($3.31 e^-/O_2$, **Figure S9**)^{8,21,24}. The deviations from the $2 e^-/O_2$ indicates the accompaniment of side reactions, supported by side products of lithium carbonate (Li_2CO_3 , 532.8 eV) and carboxylate-related bands (HCO_2Li , 531.6 eV) emerging in the TEY mode at 0.5RC and PFY mode at 0.75RC, respectively, in the O K-edge XANES spectra (**Figure 3a**). These side products result from the deteriorating carbon electrode and electrolyte through chemical reactions with superoxide intermediate during DC^{7,9} and singlet oxygen produced during RC³⁵. Deposition of the side products requires over ~4 V, reflected in the potential rise at 0.75 RC, directly after the disappearance of Li_2O_2 from the CMK-3 surface, and with CO_2 gas evolution³⁶ (**Figure 3b and 3d**). The residual side products present after RC can compromise cycling performance (**Figure S10**), which affirms the need for more stable electrolyte solution and electrode materials.

[Page 8 in SI, Figure S9] **Figure S9.** Comparison of online electrochemical mass spectrometry (OEMS) data of CMK-3 and CNT electrodes under identical conditions. CMK-3 shows $3.17 e^-/O_2$ oxidation during charge whereas the value of the same for CNT electrode is $3.31 e^-/O_2$. It is further notable that the charging overpotential for CNT is much higher, leading to larger amount of CO_2 evolution.

Comment 14: Figure S6 seems to indicate significant amounts of formate and acetate are formed during discharge, however this is not given adequate attention in the text. What is the role of parasitic products and how does it affect the interpretation of results?

Response 14: We agree with the reviewer's comment that carboxylate and carbonate signals appear together with Li_2O_2 from FT-IR after 1DC, which we have added in the revised manuscript. However, we point out that these side products are relatively minor as evidenced by negligible signal associated with side products in O K-edge XANES and also almost ideal value of $2.06 e^-/\text{O}_2$ from *in situ* pressure monitoring. From the discrepancy of FT-IR and XANES/pressure monitoring results, we also suspect the possible chemical reaction of Li_2O_2 with electrolyte molecules during sample preparation and transfer for FT-IR measurement. Since *in situ* pressure-change result confirms negligible side reactions, we believe that the effect of side product is insignificant during DC.

[Pages 5–6] Although vibration bands of side products such as lithium carboxylates and carbonates are also present in the FTIR spectra, the predominant DC product is Li_2O_2 , and is confirmed from the average electrons per oxygen molecule $2.06 e^-/\text{O}_2$, acquired by monitoring the pressure change in the $\text{Li}-\text{O}_2$ cell during DC (Figure 3b–c).

Comment 15: P. 10, line 222, how is continuous O_2 evolution indicative of a Li^+ dissolution process? The writing here was difficult to understand.

Response 15: Simply put, it can be answered that Li_2O_2 decomposition produces O_2 gas and also Li^+ coupled with electron transfer. In the revised manuscript, we have rephrased the decomposition part and our hypothesis more clearly along with the addition of the DFT results.

[Page 10] The question is then on the mechanism accounting for the facile decomposition. Since oxygen evolution reaction (OER) is a reverse process of ORR, the same DFT free energy profile in **Figure 4a** and **4b** can be used to understand the enhanced OER mechanism on the amorphous structure. The PDS for OER on both crystalline and amorphous surfaces is the reverse process of step (1), $\text{LiO}_2^* \rightleftharpoons \text{Li}^+ + \text{e}^- + \text{O}_2^*$. The η_{OER} for crystalline Li_2O_2 is calculated to be 0.86 V, which is consistent with the experimental values (~ 0.9 V, **Figure 1**). In contrast, the η_{OER} for amorphous Li_2O_2 at the most favorable binding site is calculated to be 0.48 V, almost 50 % reduction in overpotential compared to the crystalline case, consistent with experiments. With the PDS of OER being $\text{LiO}_2^* \rightleftharpoons \text{Li}^+ + \text{e}^- + \text{O}_2^*$ for both crystalline and amorphous surfaces, the underlying origin of reduced η_{OER} is again the weaker binding of $^*\text{LiO}_2$ on the amorphous surface, as in ORR, which can then be explained using the smaller number of new coordination that the adsorbed $^*\text{LiO}_2$ creates with the amorphous surface (**Figure 4c**). Interestingly, the effect of amorphous structure on the PDS of OER is much more prominent than that of ORR, explaining the great enhancement of the Li_2O_2 decomposition on the amorphous phase. To obtain further insights, Bader Population analysis for the crystalline versus amorphous surfaces before and after the $^*\text{LiO}_2$ adsorption was performed, and the results are compared in **Figure 4e-f**. It is clear that the amount of charge transferred from the surface to the adsorbed $^*\text{LiO}_2$ is indeed much more moderate on the amorphous surface due to disordered surface geometries and associated weaker electronic interactions with the adsorbates.

[Pages 11-12] The critical advantage of 1-D nanostructured Li_2O_2 on the exterior of the channel is the extensive accessibility of Li^+ in all directions to the electrolyte solution (**Scheme 1a**), unlike the case of bulk Li_2O_2 where electrolyte accessibility is restricted to the outer top surface^{38,39} (**Scheme 1b**). Such large surface area is activated by being in equilibrium with electrolyte solution (see dark blue area in **Scheme 1**) and reduces the non-active volume of inside Li_2O_2 . The vital steps of decomposition involve Li^+ dissolution and O_2 evolution which is coupled with electron and charge (hole, h^+ , in Li_2O_2) transport. With 1-D shape of Li_2O_2 , highly conductive sites are expected at the interface where Li_2O_2 , electrode and electrolyte solution closely contact (star mark in **Scheme 1a**). From these conductive sites, the charge carriers in Li_2O_2 may swiftly migrate to the activated surface where highly mobile Li^+ and charge carriers lead to facile decomposition, which can account for the decreasing height of nanostructures during RC (**Figure 2f-i**). In all, the decomposition rate and behavior of 1-D nanostructured Li_2O_2 clearly show its higher conductivity than bulk Li_2O_2 ^{40,41}. Electrochemical impedance spectroscopy (EIS) analysis in **Figure S16** demonstrates the lowest resistance of 1-D amorphous Li_2O_2 , $\sim 170 \Omega$ for the sum of interface and charge transport resistance, among other amorphous film ($\sim 220 \Omega$) and crystalline Li_2O_2 lumps ($\sim 330 \Omega$).

[Page 29, Scheme 1] **Scheme 1.** Illustration for decomposition processes of (a) 1-D nanostructured and (b) bulk-film Li_2O_2 . The dark blue region that is distinguished from inside of Li_2O_2 with the light blue color indicates activated Li_2O_2 surface where free access of Li^+ is allowed. The star mark denotes highly conductive sites where Li_2O_2 , electrode and electrolyte solution closely contact. Once RC commences, facile Li^+ dissolution and charge transport take place along to the activated Li_2O_2 surface, which results in O_2 evolution and decomposition of Li_2O_2 at low potential. The charge carrier of hole in Li_2O_2 and electron are indicated as h^+ and e^- , respectively. Since Li^+ dissolution of Li_2O_2 enclosed within the mesoporous

channel would violate charge balance rule, this part of Li_2O_2 decomposes later. The bulk Li_2O_2 film decompose sluggishly due to the limited surface area of Li_2O_2 .

[Page 12 in SI, Figure S16] **Figure S16.** Electrochemical impedance spectroscopy (EIS) analysis of (a) as-prepared and (b) 1DC CMK-3, HT-CMK-3 and CNT electrodes. The 1DC was examined at a fixed capacity of 1.0 mAh and a constant current of $50\text{ mA g}^{-1}_{\text{carbon}}$. The dotted symbols and solid lines denote experimental data and simulated curves according to (c) the equivalent circuit, respectively. The R_s includes the ohmic resistance of electrolyte solution, and the electronic resistances of current collectors and metallic Li electrode. The $R_{\text{interface}}$ and Q denote the resistance and constant phase element of an interface layer, likely a cathode electrolyte interface (CEI) layer⁴, respectively. The R_{CT} represents the charge-transfer resistance of the surface layer on the carbon electrode in parallel with its double layer capacitance, C_{dl} . The W_d is the diffusion related factors of Li^+ ion.

The $R_{\text{interface}}$ and R_{CT} values are significantly affected from the insulating Li_2O_2 after DC. From the simulated curves, the sum of $R_{\text{interface}}$ and R_{CT} for as-prepared electrodes is estimated to $84\ \Omega$ for CMK-3, $45\ \Omega$ for HT-CMK-3 and $78\ \Omega$ for CNT. It is apparent that HT-CMK-3 and CNT electrode have lower resistance than CMK-3 electrode. After DC, the sum of $R_{\text{interface}}$ and R_{CT} is increased to $170\ \Omega$ for CMK-3, $301\ \Omega$ for HT-CMK-3 and $220\ \Omega$ for CNT, demonstrating the lowest resistance of ultrathin and amorphous Li_2O_2 from CMK-3 and the highest one of crystalline and lump-shape Li_2O_2 from HT-CMK-3.

Comment 16: What is the theoretical volumetric capacity of these materials and can they be considered promising for Li-O₂ battery cathodes? What is the relevance to advance the field closer to yielding a real Li-O₂ battery?

Response 16: The theoretical volumetric capacity is $\sim 2600 \text{ mAh cm}^{-3}$ when mesoporous channels are completely filled up by Li₂O₂, corresponding to more than 6000 Wh L^{-1} volumetric energy density (at an average discharge potential of 2.6 V). We appreciate this reviewer's comment and suggestion, while we showed that the full filling of the mesoporous channels does not occur even at low current rate, thus we prefer to defer the discussion of relating this to the "real" Li-O₂ battery in the manuscript. Rather than the purpose of increasing capacity, our work has more focused on fundamental understanding of different decomposition behavior of nanostructured Li₂O₂ in comparison to bulk Li₂O₂ film, where the mesoporous carbon electrode has been used to guide the growth of one-dimensional nanostructure of Li₂O₂.

Comment 17: P. 9, lines 196-198, "This clearly verifies O₂ ingress into the mesoporous channels of CMK-3 ... as shown by higher capacitance in CV under O₂-free atmosphere." This statement is difficult to understand, please elucidate.

Response 17: We appreciate the reviewer to bringing up this point. The high capacitance from CV measured from Ar environment implies high electrochemically active surface area (ECSA), i.e., including mesoporous channels. However, this result cannot imply that the ECSA is the O₂ approaching area in the given condition. We have eliminated this part in the revised manuscript.

Comment 18: P. 9, lines 208-209, how does the total $Q_{\text{ORR}}/Q_{\text{OER}}$ ratio indicate that surface Li₂O₂ charges first?

Response 18: The total charge ratio is not directly related of the imitiation point of Li₂O₂ decomposition during RC. However, the unity of charge ratio from 1-D Li₂O₂ indicates less favorable depletion from interior of channel part, otherwise the Li₂O₂ on exterior would no longer anchor to the electrode and this physical loss would increase the charge ratio. We have rephrased this part in the revised manuscript.

[Page 11] When the total charge of ORR against OER, i.e., ($Q_{\text{ORR}}/Q_{\text{OER}}$), are estimated, they are 1.39 and 1.01 for CNT and CMK-3 electrodes, respectively. The total charge ratio at unity indicates more facile decomposition of 1-D Li_2O_2 and also the negligible loss of Li_2O_2 during OER. This also indicates the less favorable depletion of Li_2O_2 from interior of channel, which would otherwise no longer anchor the Li_2O_2 on exterior of the electrode leading the physical loss of Li_2O_2 and increase in the charge ratio.

Reviewer #2 (Remarks to the Author):

In this work, the authors show the design of lithium peroxide to one-dimensional and amorphous nanostructure using the mesoporous carbon electrode, and its pronounced effects for lithium-oxygen battery performance. This work also shows unprecedented high round-trip efficiency and fast charging for mesoporous carbon cells, which verify the rapid decomposition of nanostructured lithium peroxide during charging. Moreover, it is inferred that the enhanced battery round-trip efficiency should be attributed to the formation of amorphous Li_2O_2 , which further increases the ionic conductivity. Strong supporting data from snapshots of microscopic images, various cycling voltammetry and battery tests, and on-line electrochemical mass spectrometry confirmed the formation and the role of nanostructured lithium peroxide for fast decomposition. Overall this work is interesting and the detailed characterizations could shed light on the chemistry of $\text{Li}-\text{O}_2$ battery in different systems. I recommend its publication after addressing the following comments.

Response: We thank the reviewer for the positive remarks about this work. In the revised manuscript, we have addressed the reviewer's comments as detailed below.

Comment 1: In Figure 1, why are the discharge profiles almost the same using different carbon electrodes? Now that the charge process is greatly dependent on the carbon electrodes, why is the discharge process not affected?

Response 1: Carbon materials generally have high electronic conductivity and high activity for oxygen reduction reaction in $\text{Li}-\text{O}_2$ cells, thus the initial overpotential for discharge process is comparable with all carbon electrodes. In addition, solvated intermediate of LiO_2 and disproportionation process to form Li_2O_2 do not highly depend on electrical conductivity of Li_2O_2 in contrast to charge process. However, we also found that the overpotential from CMK-3 is 20-30 mV lower than those from other non-mesoporous carbon electrodes (**Figure S6**), due to formation of amorphous structure of Li_2O_2 . The details of DFT calculation result explaining low overpotential has been added in the revised manuscript.

[Pages 6–7] With CMK-3, the O_2 gas and electrolyte solution diffuses into the mesoporous channels by capillary force^{28, 29}, where the confined O_2 is reduced via two-electron transfer as follows:

Here, * represents adsorbed species on the surface. After step 1, Li_2O_2 can form by either step 2 or by the disproportionation of LiO_2 species ($2\text{LiO}_2 \rightleftharpoons \text{Li}_2\text{O}_2^* + \text{O}_2$). At the initial stage of DC with CMK-3, we found that the overpotential of oxygen reduction reaction (ORR), which leads to the formation of amorphous Li_2O_2 in the mesoporous channels, was lower than the crystalline Li_2O_2 formed with the non-mesoporous electrodes (**Figure S6**). The lower ORR overpotential on the amorphous Li_2O_2 surface can be understood by using density functional theory (DFT) calculations on the key intermediates of the electrochemical reactions.

The DFT calculations included two cycles of Li_2O_2 formation since the calculation unit cell contains two formula units of Li_2O_2 product. **Figure 4a** shows the calculated free energy diagram along with the

optimized structures at three different applied potentials (U). The free energy of O_2 was indirectly calculated according to the water-splitting reaction, and the energy of ($Li^+ + e^-$) is assumed to be in equilibrium with the bulk Li, in a way similar to the Computational Hydrogen Electrode method³⁰. The applied potential (U) is incorporated into the energetics *via* $-neU$, where n is the number of transferred ($Li^+ + e^-$). The calculated overpotential (η) for ORR is defined as $\eta_{ORR} = U_0 - U_{DC}$ where U_0 and U_{DC} are the equilibrium and DC potentials, respectively^{31,32}. Using these prescriptions, the η_{ORR} for the crystalline structure, reconstructed Li_2O_2 ($1\bar{1}00$) that is the most stable surface³¹, is calculated to be 0.52V, which is in good agreement with the experimental value ($\eta_{ORR} = 0.4 - 0.5$ V, from U_0 of 3.15 V, **Figure 1** and **Figure S6**). For the amorphous Li_2O_2 surface, the structure suggested from Tian et al²³ determined from first-principles molecular dynamics simulations was utilized. We divided the surface into 16 equivalent areas to find the lowest energy adsorption site (**Figure S7**), and the free energy diagram shown in **Figure 4b** is based on the latter most favorable reaction site. However, we note as summarized in **Table S2**, that statistically 14 sites out of 16 still show lower η_{ORR} compared to that from the crystalline surface, which is in accordance with the experimental results (**Figure S6**).

For both amorphous and crystalline surfaces, the potential determining step (PDS) for ORR is calculated to be step (2), indicating that the weaker binding of second $*LiO_2$ adsorption on the amorphous surface is responsible for a lower η_{ORR} . Structurally, this weaker binding of $*LiO_2$ on the amorphous surface is due to the disordered arrangements of the surface Li and O atoms, preventing the newly adsorbing $*LiO_2$ species from forming optimal coordination with the surface. To quantify this, in **Figure 4f**, we plotted the binding energies of $*LiO_2$ for the PDS of ORR as a function of the number of newly formed Li-O bonds as a result of $*LiO_2$ adsorption. Overall, the low $*LiO_2$ binding energies on the amorphous surface are indeed associated with the low coordination number that the newly adsorbed $*LiO_2$ forms with the surface, compared to that on the crystalline surface.

Figure 4. Density functional calculations and Bader charge analysis. (a–b) Calculated free energy diagrams (a) on the Li_2O_2 ($1\bar{1}00$) and (b) on amorphous Li_2O_2 , along with the optimized structures. For the amorphous surface, the most favorable adsorption site was considered to construct the diagram. Lithium is colored in light gray (bulk) and light green (adsorbate); Oxygen is colored in dark gray (bulk) and red (adsorbate). (c–d) The variation in electron density upon the first LiO_2 adsorption (e) on the crystalline ($1\bar{1}00$), and (f) amorphous Li_2O_2 surfaces. The charge density isosurfaces contour is plotted with a threshold value of $0.01e \text{ \AA}^{-3}$. The newly adsorbing *LiO_2 is denoted explicitly as Li, O_1 , and O_2 . O_2 which is not shown in (c) is behind O_1 . Lithium is in light green and oxygen in red. (e–f) Binding energies of 1st LiO_2 (e) and 2nd LiO_2 (f) for 16 sites of the amorphous Li_2O_2 surface plotted versus coordination number. Binding energies on the crystalline surface are also shown for comparison, denoted as c- Li_2O_2 .

Comment 2: It is appealing to achieve enhanced round-trip efficiency thanks to the formation of amorphous 1-D Li_2O_2 . Would the amorphous Li_2O_2 still be generated at high discharge rate? If so, how to improve it? Relevant data or insightful discussion need be added.

Response 2: We appreciate the reviewer to bringing up this point. The additional XRD data for Li_2O_2 film formed at the high DC rate of 100 mA g^{-1} shows negligible crystalline reflections, confirming the amorphous structure. However, there are morphology changes at high DC current rates from 1-D nanostructure to film, which causes more sluggish decomposition. To promote facile Li_2O_2 decomposition, the mesoporous pore size needs to be tuned to enable the formation of nanostructured Li_2O_2 to also occur at higher DC rate. In the revised manuscript, we added XRD data of Li_2O_2 at 100 mA g^{-1} . However, we decided to defer discussions on the engineering the CMK-3 electrode to further improve rate capability, which is beyond of the scope in our work, which focuses on more fundamental understanding of facile decomposition with nanostructured Li_2O_2 .

[Page 9] Both flake and film of Li_2O_2 have amorphous structure (**Figure S11**), thus their different shapes are strongly correlated with decomposition behaviors that are explored with anodic linear sweep voltammetry (LSV) (**Figure 5b**).

[Page 9 in SI, Figure S11] **Figure S11.** XRD patterns of DC product of conformal film acquired at a current rate of $100 \text{ mA g}^{-1}_{\text{carbon}}$ and a fixed DC capacity of $500 \text{ mAh g}^{-1}_{\text{carbon}}$.

Comment 3: The authors claimed that amorphous Li_2O_2 give rise to increased ionic conductivity. This point can be made more convincing with more evidences by EIS test or other characterization. And the authors are suggested to provide more systematic cycling performance.

Response 3: We appreciate the reviewer to bringing up this point. We have measured EIS, which revealed the highest conductivity of Li_2O_2 formed with CMK-3 in comparison to CNT and high-temperature treated CMK-3. Along with the EIS data, we also have added the cycling DC-RC profiles, which show the gradual increase in RC potential.

[Pages 11–12] Electrochemical impedance spectroscopy (EIS) analysis in **Figure S16** demonstrates the lowest resistance of 1-D amorphous Li_2O_2 , $\sim 170 \Omega$ for the sum of interface and charge transport resistance, among other amorphous film ($\sim 220 \Omega$) and crystalline Li_2O_2 lumps ($\sim 330 \Omega$).

[Page 12 in SI, Figure S16] **Figure S16.** Electrochemical impedance spectroscopy (EIS) analysis of (a) as-prepared and (b) 1DC CMK-3, HT-CMK-3 and CNT electrodes. The 1DC was examined at a fixed capacity of 1.0 mAh and a constant current of $50 \text{ mA g}^{-1}_{\text{carbon}}$. The dotted symbols and solid lines denote experimental data and simulated curves according to (c) the equivalent circuit, respectively. The R_s includes the ohmic resistance of electrolyte solution, and the electronic resistances of current collectors and metallic Li electrode. The $R_{\text{interface}}$ and Q denote the resistance and constant phase element of an interface layer, likely a cathode electrolyte interface (CEI) layer⁴, respectively. The R_{CT} represents the charge-transfer resistance of the surface layer on the carbon electrode in parallel with its double layer capacitance, C_{dl} . The W_d is the diffusion related factors of Li^+ ion.

The $R_{\text{interface}}$ and R_{CT} values are significantly affected from the insulating Li_2O_2 after DC. From the simulated curves, the sum of $R_{\text{interface}}$ and R_{CT} for as-prepared electrodes is estimated to 84Ω for CMK-3, 45Ω for HT-CMK-3 and 78Ω for CNT. It is apparent that HT-CMK-3 and CNT electrode have lower resistance than CMK-3 electrode. After DC, the sum of $R_{\text{interface}}$ and R_{CT} is increased to 170Ω for CMK-3, 301Ω for HT-CMK-3 and 220Ω for CNT, demonstrating the lowest resistance of ultrathin and amorphous Li_2O_2 from CMK-3 and the highest one of crystalline and lump-shape Li_2O_2 from HT-CMK-3.

[Page 8 in SI, Figure S10] **Figure S10.** Cycling performance of CMK-3. (a) The round-trip efficiency and (b) DC-RC curves at a limited capacity of $500 \text{ mAh g}^{-1}_{\text{carbon}}$ and current rate of $50 \text{ mA g}^{-1}_{\text{carbon}}$.

Reviewers' comments:

Reviewer #1 (Remarks to the Author):

The authors have done a very thorough job of addressing my original questions, and the modifications greatly improve the manuscript. I only have one additional question, which pertains to the added QM calculations of the discharge and charge voltages. The authors define an overpotential on p. 7, $U_o - UDC$, which they compare to experimentally observed values and find good agreement. My concern is with the choice of U_o , which the authors take as 3.15 V. This is higher than the thermodynamic potential of Li_2O_2 formation (2.96 V). Where does it come from? Since the authors reference Figure 1, it appears that this might come from the OCV? If so, this may not be accurate because it is not clear what the OCV before discharge really reflects in Li-O₂ cells. It might simply reflect the surface state of carbon with its own surface contamination, prior to discharge. Viswanathan et al. (J. Phys. Chem. Lett. 2013, 4, 556–560) and Li et al. (Energy Environ. Sci., 2015, 8, 182) independently reported that the OCV of carbon electrodes with some Li_2O_2 formed on it is actually ~ 2.85 V. So, the authors may want to reconsider how they present these findings, especially because the OCV prior to discharge in their cells would not contain any Li_2O_2 (while their calculations do). It seems to me that the computational description on ORR is less important to this overall story and perhaps this part could be omitted without loss of clarity. However, the authors should perhaps soften or refrain from quantitative comparisons of overpotential on charge.

Reviewer #2 (Remarks to the Author):

The authors have very carefully addressed the reviewers' comments. I find their responses satisfactory, and therefore recommend its acceptance.

Reviewer #1 (Remarks to the Author):

The authors have done a very thorough job of addressing my original questions, and the modifications greatly improve the manuscript. I only have one additional question, which pertains to the added QM calculations of the discharge and charge voltages. The authors define an overpotential on p. 7, $U_0 - U_{DC}$, which they compare to experimentally observed values and find good agreement. My concern is with the choice of U_0 , which the authors take as 3.15 V. This is higher than the thermodynamic potential of Li_2O_2 formation (2.96 V). Where does it come from? Since the authors reference Figure 1, it appears that this might come from the OCV? If so, this may not be accurate because it is not clear what the OCV before discharge really reflects in Li-O₂ cells. It might simply reflect the surface state of carbon with its own surface contamination, prior to discharge. Viswanathan et al. (*J. Phys. Chem. Lett.* 2013, 4, 556–560) and Li et al. (*Energy Environ. Sci.*, 2015, 8, 182) independently reported that the OCV of carbon electrodes with some Li_2O_2 formed on it is actually ~ 2.85 V. So, the authors may want to reconsider how they present these findings, especially because the OCV prior to discharge in their cells would not contain any Li_2O_2 (while their calculations do). It seems to me that the computational description on ORR is less important to this overall story and perhaps this part could be omitted without loss of clarity. However, the authors should perhaps soften or refrain from quantitative comparisons of overpotential on charge.

Response: We thank the reviewer for the positive comments regarding the revised manuscript. We do agree with the reviewer that the equilibrium potential of Li_2O_2 formation is 2.96 V. The 3.15 V that the reviewer is referring to is the calculated value and not from the current experimental data. However, re-reading the original sentence of our manuscript, we do see a possibility of misunderstanding, and so to clarify it, we have revised that part on pages 6–7 as: “The overpotential (η) for ORR is defined as $\eta_{\text{ORR}} = U_0 - U_{\text{DC}}$, where the equilibrium potential of U_0 is defined as the potential for which the change of free energy for the whole process is zero, and the DC potential of U_{DC} is the highest potential that makes the free energy for every step in ORR, downhill.^{30,31} Using these prescriptions, the calculated $U_0 = 3.15$ V for the crystalline structure, reconstructed Li_2O_2 (1 $\bar{1}$ 00) that is the most stable surface³⁰, is in good agreement with the experimental thermodynamic potential of Li_2O_2 formation (2.96 V).³²”

Ref[32] :Girishkumar, G., McCloskey, B., Luntz, A. C., Swanson, S. & Wilcke, W. Lithium–Air Battery: Promise and Challenges. *J. Phys. Chem. Lett.* **1**, 2193-2203, (2010).

We also fully agree with the reviewer that the computational description regarding ORR is not significantly necessary and have moved this part to Methods in the revised manuscript.